# MULTI-TASK CORRUPTED PREDICTION FOR LEARNING ROBUST AUDIO-VISUAL SPEECH REPRESENTATION

**Sungnyun Kim**[1], **Sungwoo Cho**[1], **Sangmin Bae**[1], **Kangwook Jang**[2], **Se-Young Yun**[1]
[1]Kim Jaechul Graduate School of AI, KAIST
[2]School of Electrical Engineering, KAIST

## ABSTRACT

Audio-visual speech recognition (AVSR) incorporates auditory and visual modalities to improve recognition accuracy, particularly in noisy environments where audio-only speech systems are insufficient. While previous research has largely addressed audio disruptions, few studies have dealt with visual corruptions, *e.g.,* lip occlusions or blurred videos, which are also detrimental. To address this real-world challenge, we propose CAV2vec, a novel self-supervised speech representation learning framework particularly designed to handle audio-visual joint corruption. CAV2vec employs a self-distillation approach with a corrupted prediction task, where the student model learns to predict clean targets, generated by the teacher model, with corrupted input frames. Specifically, we suggest a *unimodal multi-task learning*, which distills cross-modal knowledge and aligns the corrupted modalities, by predicting clean audio targets with corrupted videos, and clean video targets with corrupted audios. This strategy mitigates the dispersion in the representation space caused by corrupted modalities, leading to more reliable and robust audio-visual fusion. Our experiments on robust AVSR benchmarks demonstrate that the corrupted representation learning method significantly enhances recognition accuracy across generalized environments involving various types of corruption. Our code is available at `https://github.com/sungnyun/cav2vec`.

## 1 INTRODUCTION

Audio-visual speech recognition (AVSR) (Noda et al., 2015; Afouras et al., 2018a; Ma et al., 2021b; Shi et al., 2022a; Hsu & Shi, 2022; Hu et al., 2023a) represents a significant advancement in speech recognition by integrating both auditory and visual modalities to enhance performance. This multimodal integration proves particularly vital in contexts where audio-only speech recognition systems suffer from ambient or background noise, as visual speech information like lip movements significantly improves recognition capabilities (Makino et al., 2019; Ren et al., 2021; Chen et al., 2023). In this sense, previous works on AVSR have primarily focused on overcoming audio disruptions, *e.g.,* Xu et al. (2020) training an audio enhancement sub-network, or Shi et al. (2022b) pretraining with noise-augmented audio. Nonetheless, real-world applications often encounter scenarios where video corruption is as critical as audio disturbances. For example, we may consider an outdoor interview where not only is the audio disturbed by ambient noise or traffic sound but visual cues are also intermittently occluded, either when the speaker's hands obstruct the view of their face or a camera is out of focus (see Figure 1a). AVSR models often fail to accurately recognize the utterance under these corrupted environments.

Despite the effectiveness of current methods in addressing audio corruptions, there is a lack of solutions for visual corruption, highlighting the necessity for a more robust approach to tackle audio-visual joint corruption in AVSR systems. Recent efforts have addressed the visual corruption using techniques such as scoring modules to assess the reliability of audio and video frames (Hong et al., 2023), or generative pipelines to reconstruct occluded face images (Wang et al., 2024). However, these methods often rely on specific architectures or external modules, which limit their applicability. Building on recent advances in audio-visual self-supervised learning that highlight the efficacy of modality-fusion representations (Shi et al., 2022a; Lian et al., 2023; Zhang et al., 2023), we propose **CAV2vec**, a novel audio-visual speech representation learning method designed to handle jointly **c**orrupted **a**udio-**v**isual data. CAV2vec is trained through a *corrupted prediction task*, where the model

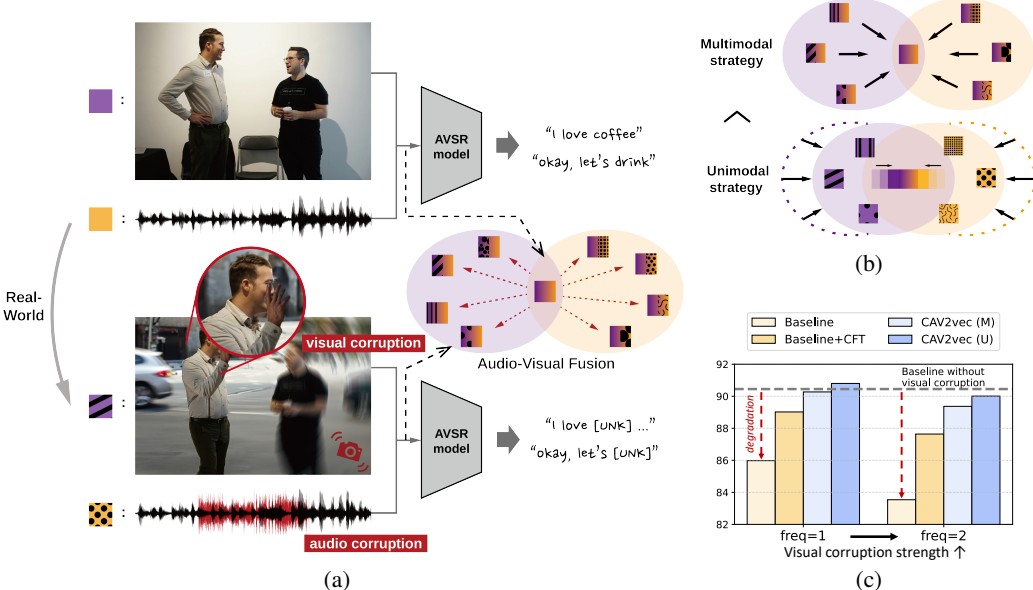

Figure 1: (a) Real-world speech recognition challenges. AVSR models suffer from maintaining robust representations under the corrupted environments and fail to recognize utterances. (b) Our corrupted representation learning strategies with multimodal and unimodal corrupted prediction tasks. (c) Speech recognition accuracy ($100 - $ WER %), where frequency denotes the number of visual corruption events in a sequence. Our representation learning framework, CAV2vec with a unimodal strategy (U), significantly improves robustness compared to the baseline model and even outperforms the multimodal strategy (M).

learns to predict clean targets from corrupted input sequences. For this, we employ a teacher-student self-distillation framework (Caron et al., 2021; Ruan et al., 2023), which has proven effective in learning contextualized speech representations (Baevski et al., 2020; 2022; Shi et al., 2022b; Zhu et al., 2024; Liu et al., 2024) without requiring architectural changes or additional modules. In this framework, corrupted sequences are fed into the student model, while the self-evolving teacher model generates the clean targets online.

Within the CAV2vec representation learning framework, the corrupted prediction task can be defined in a multimodal or unimodal strategy. A multimodal strategy, inspired by the masked prediction task in AV-data2vec (Lian et al., 2023), involves corrupting both audio and video inputs to generate corrupted multimodal features, with the model learning to predict clean multimodal targets. While this approach enhances the robustness of multimodal representations, it is less capable of isolating the effects of corruption on individual modalities, as both inputs and targets contain mixed audio-visual information. Multimodal fusion is often prone to combining redundant information (Hsu & Shi, 2022; Mai et al., 2023), where discriminative unimodal information is ignored, thereby the multimodal prediction falls into overfitting. Previous approaches have tackled this by improving cross-modal information, such as Mai et al. (2023) applying late-fusion to filter out noisy information from unimodal features, and Hu et al. (2023b) learning a viseme-phoneme mapping (Bear & Harvey, 2017) to restore corrupted phonemes through visemes, but both of them rely on the information bottleneck structure before the fusion.

To address the limitations of multimodal prediction approach, we introduce CAV2vec with a *unimodal multi-task learning strategy* for corrupted prediction tasks, which leverages corrupted unimodal sequences to distill cross-modal knowledge. Our unimodal strategy involves predicting clean audio targets with corrupted videos and predicting clean video targets with corrupted audios. In AVSR, this cross-modal alignment (Ren et al., 2021; Hu et al., 2023b;c) is essential for effectively integrating information from both modalities. Our unimodal prediction strategy, as depicted in Figure 1b, improves cross-modal alignment by reducing the dispersion in representation caused by the corrupted inputs. Figure 1c describes the AVSR performance under audio-visual jointly corrupted environments. While fine-tuning with corrupted data (CFT) improves robustness to some extent, it remains challenging at higher degrees of corruption. By incorporating the corrupted representation learning before CFT, CAV2vec demonstrates superior performance. CAV2vec with unimodal multi-task learning further enhances the corrupted prediction framework, effectively aligning the corrupted modalities and achieving more robust results. We summarize our contributions as follows.

- ○ We propose a novel audio-visual speech representation learning, CAV2vec, specifically designed for robustness under audio-visual joint corruption within a self-distillation framework.

- ○ CAV2vec conducts a unimodal multi-task learning for corrupted prediction tasks, predicting clean targets from corrupted input sequences. This strategy effectively enhances cross-modal alignment between corrupted audio and video for reliable multimodal fusion.

- ○ We establish an AVSR benchmark for generalization so that our setup includes novel types of corruption unseen during training, *e.g.,* mouth occlusion by hands or face pixelation along with public noise, allowing for diverse assessment of model robustness to audio-visual joint corruption. CAV2vec demonstrates significant performance improvement on our robust AVSR benchmarks.

## 2 RELATED WORK

**Audio-visual speech recognition models**  Automatic speech recognition (ASR) methods that transcribe speech audio to text have been extensively studied for years (Schneider et al., 2019; Gulati et al., 2020; Baevski et al., 2020; Hsu et al., 2021; Chen et al., 2022; Chiu et al., 2022). However, since audio signals are often disrupted by background noise, multimodality, particularly visual information from speech video, has been incorportaed into the ASR system (Makino et al., 2019; Pan et al., 2022; Shi et al., 2022a; Seo et al., 2023; Ma et al., 2023). In these multimodal approaches, the audio modality captures the acoustic features of speech signal, while the video modality provides visual information about the speaker's face and lip movements. This integration enhances the robustness of the ASR system and improves its performance, especially in noisy audio environments.

Several studies aimed at aligning and jointly training audio-visual modalities have been developed in an end-to-end learning framework (Dupont & Luettin, 2000; Ma et al., 2021b; Hong et al., 2022; Burchi & Timofte, 2023) or self-supervised pretraining approach (Ma et al., 2021a; Qu et al., 2022; Shi et al., 2022a; Seo et al., 2023; Zhu et al., 2023) that often utilizes masked modeling of speech representations. Recently, among the multimodal speech recognition models leveraging the self-supervised learning, a self-distillation approach (Lian et al., 2023; Haliassos et al., 2023; 2024; Zhang et al., 2024b; Liu et al., 2024), which learns the contextualized representations by distilling the self-evolving teacher's knowledge for masked inputs, has demonstrated superior performances.

**Learning robustness for AVSR**  The AVSR studies have focused on developing robust models for various types of noise while simultaneously utilizing audio and visual information. Most of this research has initially focused on addressing noise in the audio modality (Shi et al., 2022b; Hu et al., 2023b;c; Chen et al., 2023; Kim et al., 2024; Ithal et al., 2024). However, more recent efforts have explored disturbances in visual data, creating robust models by adding background noise, such as Gaussian noise, to the speech videos. In this line of research, Hong et al. (2023) have considered that human speech often involves a mouth region being occluded by objects and first applied visual occlusion into speech data. This has inspired further research addressing similar challenges by restoring the occluded images by a generative model (Wang et al., 2024). Additionally, Fu et al. (2024) have combined prompt learning with contrastive learning to deal with audio-visual asynchrony, while Zhang et al. (2024a) have examined the issue of a completely missing visual modality, generating the visual hallucination during inference. Li et al. (2024) demonstrate the effectiveness of leveraging unified cross-modal attention and a synchronization module to encode audio and video sequences in a unified feature space. In our study, we address real-world challenges of audio-visual joint corruption by using corrupted representation learning, offering a general framework without relying on specific architectures or external modules.

## 3 PRELIMINARIES

### 3.1 NOTATIONS

Let $A = [a_1, a_2, \ldots, a_T]$ be the audio sequence and $V = [v_1, v_2, \ldots, v_T]$ be the video sequence over time $T$. We define a set of corrupted indices for the audio sequence as $C^a \subseteq \{1, 2, \ldots, T\}$ and a set of corrupted indices for the video sequence as $C^v \subseteq \{1, 2, \ldots, T\}$. To model the corruption, we define a family of corruption functions $\Omega$ that can transform or corrupt certain frames of data. Thus, given some corruption functions $\omega^a, \omega^v \in \Omega$, the corrupted audio sequence $\tilde{A} = [\tilde{a}_1, \tilde{a}_2, \ldots, \tilde{a}_T]$

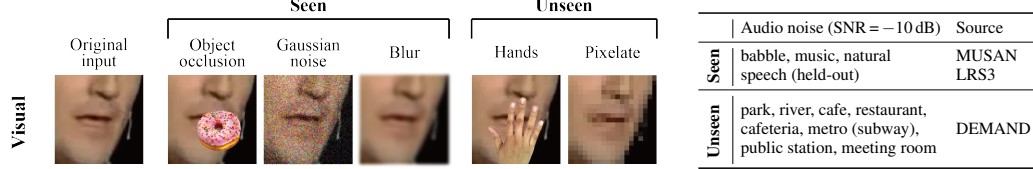

Figure 2: The visual and audio corruption types we use in our training and evaluation phases. Unseen corruption types are only utilized in evaluation to assess the model's generalizability. The speech audio noise from LRS3 is ensured that there is no speaker overlap between train and evaluation sets.

and the corrupted video sequence $\tilde{V} = [\tilde{v}_1, \tilde{v}_2, \ldots, \tilde{v}_T]$ are defined as follows.

$$\tilde{a}_t = \begin{cases} \omega^a(a_t) & \text{if } t \in C^a \\ a_t & \text{else} \end{cases} \quad \text{and} \quad \tilde{v}_t = \begin{cases} \omega^v(v_t) & \text{if } t \in C^v \\ v_t & \text{else} \end{cases} \tag{1}$$

Raw audio and video data are processed by its respective feature extractor, and the resulting features are concatenated before being input to the multimodal Transformer encoder $f_\theta : \mathbb{R}^{T \times D} \to \mathbb{R}^{T \times D}$. The output feature sequence for this multimodal input is denoted as $\tilde{Z}^{av} = f_\theta(\tilde{A}; \tilde{V}) = [\tilde{z}_1^{av}, \tilde{z}_2^{av}, \ldots, \tilde{z}_T^{av}]$. If $t \in C^a \cup C^v$, then $\tilde{z}_t^{av}$ is considered a corrupted feature representation.

## 3.2 MASKED PREDICTION TASK

**Self-distillation framework** The self-distillation approach as self-supervised learning has been shown to be highly effective in learning contextualized representations (Baevski et al., 2022; 2023; Liu et al., 2024) without supervised labels, including in multimodal feature spaces (Zhang et al., 2023; 2024b; Zhu et al., 2024). In this framework, a reference function $f$, commonly referred to as the teacher model, is updated by an exponential moving average (EMA) of the parameterized student model $f_\theta$ with a decaying parameter $\eta$, $f \leftarrow \eta * f + (1 - \eta) * f_\theta$. The student model learns by predicting targets generated online by the teacher. Thus, it enables representation learning without requiring external modules or modifications to the overall model structure.

**Masked prediction task loss** In AV-data2vec (Lian et al., 2023), audio and video frames are randomly masked, and a masked prediction task is performed by predicting each masked frame with the target feature. The target features are obtained from clean, unmasked data using the teacher model. Then, the masked prediction loss is defined as:

$$\mathcal{L}_{\text{MASK}} = \sum_{t \in M^a \cup M^v} \ell\Big([f_\theta(\text{MASK}(A); \text{MASK}(V))]_t, \ [f(A; V)]_t\Big) \tag{2}$$

where $M^a$ and $M^v$ are the set of indices of masked audio and video frames, respectively. $\ell$ is often used as a mean squared error (MSE) loss, and $f(\cdot)$ as the teacher model's average representation—the output sequence averaged over top-$k$ Transformer blocks to establish the target, *i.e.,* $f(A; V) = \frac{1}{k} \sum_{l=L-k+1}^{L} f^l(A; V)$, where $L$ is the number of blocks.

## 4 CAV2VEC: UNIMODAL MULTI-TASK CORRUPTED PREDICTION

### 4.1 VISUAL AND AUDIO CORRUPTION TYPES

Figure 2 presents the visual and audio corruption types used in this study. We propose a novel evaluation benchmark, introducing corruption types unseen during training, to assess the model's generalizability under diverse and realistic conditions. During training, visual corruptions include object occlusion, Gaussian noise, and blurring, applied to video frames following Hong et al. (2023). For object occlusion, we obscure the mouth regions by COCO (Lin et al., 2014) object images, as presented in Voo et al. (2022). In evaluation, we apply unseen visual corruption types, using the 11k-Hands dataset (Afifi, 2019) to occlude the mouth regions with diverse hand images. Furthermore, we apply corruption by pixelating the entire frame using the `opencv` library, with a 3x3 patch interpolation. This allows us to evaluate the model's robustness under conditions where human faces are often pixelated due to privacy or ethical concerns.

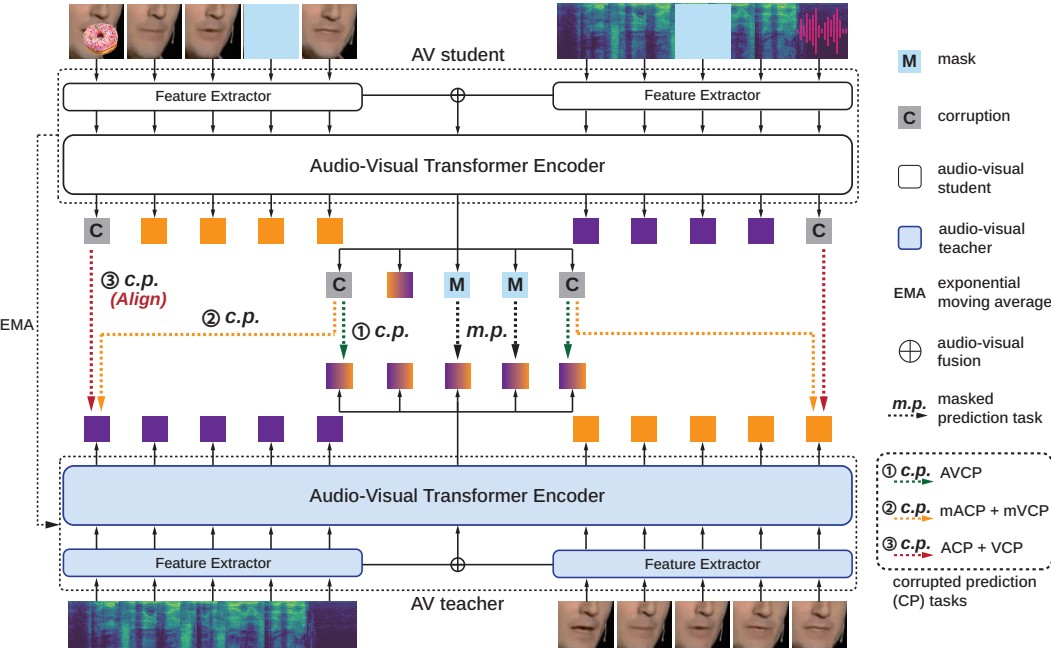

Figure 3: Overview of our representation learning framework with corrupted prediction tasks. For the corrupted prediction strategies, focusing on the cross-modal alignment through unimodal multi-task learning proves highly effective in gaining multimodal robustness.

For the audio corruption, we use various types of background noise that are recorded from different sources, at a $-10\,\mathrm{dB}$ SNR (signal-to-noise ratio). In the training, we apply conventionally used audio noise types (Shi et al., 2022b; Hsu & Shi, 2022), babble, music, and natural noise sampled from MUSAN (Snyder et al., 2015), and speech noise sampled from LRS3 (Afouras et al., 2018b), onto the original speech signal. In the evaluation, we introduce new corruption types from the DEMAND dataset (Thiemann et al., 2013), which includes real-world indoor and outdoor recordings. Out of 18 categories of recording, we evaluate on 8 relatively noisy environments: park, river, cafe, restaurant, cafeteria, metro, public station, and meeting room. For results in the remaining categories, *e.g.,* living room or office, refer to Appendix C.2.

## 4.2 CORRUPTED PREDICTION TASKS OF CAV2VEC

We present **CAV2vec**, a robust representation learning framework to account for corrupted audio-visual sequences. Inspired by the idea of conventional masked prediction strategy (Lian et al., 2023), CAV2vec is trained through a *corrupted prediction task*. The corrupted prediction task loss is designed to minimize the difference between the student model's output for the corrupted data and the teacher model's output for the uncorrupted data. Figure 3 provides an overview of the corrupted representation learning of CAV2vec. We apply corruption functions to both video and audio data (see Figure 2) and perform the corrupted prediction tasks on the corrupted frames, alongside the masked prediction task on the masked frames. We suggest different strategies in designing these corrupted prediction tasks, depending on the modality of input and target representations.

**Audio-visual corrupted prediction task**   A straightforward approach is using corrupted multimodal inputs as well as clean multimodal targets (① in Figure 3), following the masked prediction strategy. We name it as audio-visual corrupted prediction (AVCP) task, where the AVCP loss function is analogously defined as:

$$\mathcal{L}_{\text{AVCP}} = \sum_{t \in C^a \cup C^v} \ell\Big(\tilde{z}_t^{av}, [f(A; V)]_t\Big). \tag{3}$$

Here, the loss is computed only for the corrupted indices $t \in C^a \cup C^v$, where $\tilde{Z}^{av}$ represents the student model's predictions for (possibly) corrupted sequences $\tilde{A}$ and $\tilde{V}$, and $f(A; V)$ represents the target for clean sequences $A$ and $V$.

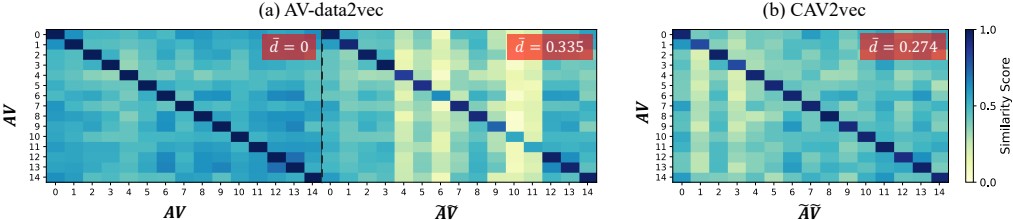

Figure 4: Similarity scores measured between audio-visual features of sample sequences. Clean sequence representations are compared with corrupted ones from (a) AV-data2vec and (b) CAV2vec. The normalized L2 distance $\bar{d}$ is calculated between the clean and corrupted features per-sample.

**Unimodal corrupted prediction tasks**  While the AVCP task in Eq. (3) is effective in learning robustness for multimodal features, the mixed audio-visual information in both inputs and targets makes it hard to isolate corruptions on individual modalities (Mai et al., 2023). To address this, we introduce CAV2vec with a *unimodal multi-task learning strategy* for corrupted prediction, leveraging audio-only and video-only sequences (③ in Figure 3). These unimodal tasks enhance cross-modal alignment, which is essential in AVSR for capturing multimodal correlations (Ren et al., 2021; Hu et al., 2023b;c), particularly when corruption disrupts the link between two modalities. We propose the unimodal tasks to distill cross-modal knowledge: audio corrupted prediction (ACP) task and visual corrupted prediction (VCP) task. Their loss functions are defined as:

$$\mathcal{L}_{\text{ACP}} = \sum_{t \in C^v} \ell\Big(\tilde{z}_t^v, \, [f(A; \mathbf{0})]_t\Big), \quad \mathcal{L}_{\text{VCP}} = \sum_{t \in C^a} \ell\Big(\tilde{z}_t^a, \, [f(\mathbf{0}; V)]_t\Big) \tag{4}$$

where $\tilde{Z}^v = f_\theta(\mathbf{0}; \tilde{V})$ and $\tilde{Z}^a = f_\theta(\tilde{A}; \mathbf{0})$ denote video-only and audio-only unimodal features, respectively. Thus, the ACP task predicts clean audio targets with corrupted videos, and the VCP task predicts clean video targets with corrupted audios. To thoroughly examine the impact of each task and the relationship with modality alignment, we also implement multimodal ACP and VCP tasks, called mACP and mVCP, which use multimodal inputs $\tilde{Z}^{av}$ and unimodal targets (② in Figure 3). These are formulated as $\mathcal{L}_{\text{mACP}} = \sum_{t \in C^v} \ell(\tilde{z}_t^{av}, [f(A; \mathbf{0})]_t)$ and $\mathcal{L}_{\text{mVCP}} = \sum_{t \in C^a} \ell(\tilde{z}_t^{av}, [f(\mathbf{0}; V)]_t)$, and their effectiveness is investigated in Section 5.3.

**Overall multi-task loss of CAV2vec**  We employ both corrupted prediction in Eq. (4) and masked prediction in Eq. (2), but we do not allow the overlap between masked and corrupted frames to separate the tasks, *i.e.*, $(M^a \cup M^v) \cap (C^a \cup C^v) = \emptyset$. We have empirically found this task separation to be effective. Incorporating the unimodal corrupted prediction tasks with modality dropout (Hsu & Shi, 2022) as well as the masked prediction task for multimodal inputs, the loss function for CAV2vec within multi-task learning is defined as follows:

$$\mathcal{L}_{\text{CAV2vec}} = \lambda_{\text{ACP}} \mathcal{L}_{\text{ACP}} + \lambda_{\text{VCP}} \mathcal{L}_{\text{VCP}} + \lambda_{\text{MASK}} \mathcal{L}_{\text{MASK}} + \lambda_{\text{MLM}} \mathcal{L}_{\text{MLM}} \tag{5}$$

where $\mathcal{L}_{\text{MLM}}$ is a masked language modeling-style (MLM) loss used in Shi et al. (2022a); Zhang et al. (2023). $\mathcal{L}_{\text{MLM}}$, which predicts the cluster index of the masked features in a cross-entropy loss form, helps the model converge faster than using $\mathcal{L}_{\text{MASK}}$ alone (Zhang et al., 2023). In our experiments, we set $\lambda_{\text{ACP}} = \lambda_{\text{VCP}} = \lambda_{\text{MASK}} = 1.0$ and $\lambda_{\text{MLM}} = 2.0$ to match the scales of each loss.

As shown in Figure 4a, corruption disrupts audio-visual representations, resulting in reduced similarity scores and increased dispersion of features, which hinders the model's ability to maintain robust representations. In Figure 4b, our corrupted representation learning helps the model encode highly correlated audio-visual representations, restoring high similarity (small distance) between clean and corrupted sequence features and leading to a more compact and resilient representation space.

## 5 EXPERIMENTS AND RESULTS

### 5.1 IMPLEMENTATION DETAILS

**Datasets**  We train and evaluate our model on LRS3 (Afouras et al., 2018b), which contains roughly 433 hours of TED talks from over 5,000 speakers. Most of our experimental configurations follow Shi et al. (2022b), including noise augmentation and evaluation protocols. Audio noise is extracted from

the MUSAN (Snyder et al., 2015) (*babble*, *music*, *natural*) and LRS3 (*speech*) datasets, partitioned into training, validation, and test sets. This noise is added to audio waveform during both training and evaluation. The evaluation metric for AVSR is the word error rate (WER, %). For audio feature extraction, we extract 26-dimensional log filter bank features from raw audio at a stride of 10 ms, stacking 4 adjacent frames to achieve a frame rate of 25 fps. Video track is sampled at 25Hz, with a 96×96 region center-cropped on the speaker's mouth. During training, we randomly crop an 88×88 region and apply horizontal flips with probability 50%.

**CAV2vec and baseline models** CAV2vec uses 24 Transformer (Vaswani et al., 2017) block layers for the AVSR encoder and 9 layers for the decoder, based on the AV-HuBERT-LARGE model (Shi et al., 2022a). The visual feature extractor is a modified ResNet-18, while the audio feature extractor is a linear projection layer. The extracted features are concatenated to form fusion audio-visual features, which are input to the Transformer encoder. We initialize the model using the publicly available checkpoint from Shi et al. (2022b), the AV-HuBERT encoder pretrained on noise-augmented LRS3 (Afouras et al., 2018b) + VoxCeleb2 (Chung et al., 2018), and proceed our representation learning phase afterwards. This training strategy makes the whole process efficient, spanning only 2% of AV-HuBERT's pretraining cost (Shi et al., 2022a), as well as leveraging high-resource knowledge of VoxCeleb2 (1,326 hours). Our representation learning can thus be considered as an uptraining phase (Ainslie et al., 2023), an additional pretraining step that helps the model adapt to corrupted data before fine-tuning on the supervised speech recognition task. As in prior self-distillation works (Caron et al., 2021; Baevski et al., 2022), we employ single MLP-layer predictors assigned for each task on the student model, which are removed after training.

For baseline models, we compare with (1) V-CAFE (Hong et al., 2022), (2) RAVEn (Haliassos et al., 2023), (3) BRAVEn (Haliassos et al., 2024), (4) AV-HuBERT (Shi et al., 2022a), (5) AV-data2vec (Lian et al., 2023), and (6) AV-RelScore (Hong et al., 2023). V-CAFE is an end-to-end supervised model with a relatively small model size, while the other models are pretrained with their own loss functions. We re-implemented RelScore on the AV-HuBERT backbone, as the original version based on V-CAFE (Hong et al., 2023) performs poorly, and further trained the scoring module. All baseline models are initialized from their respective pretrained checkpoints and then fine-tuned using the same decoder architecture and training configurations as our model. For details of each baseline model, refer to Appendix A.1.

**Training configuration** For CAV2vec uptraining, we sample audio noise at 0 dB SNR and randomly perturb the clean speech signal with a 25% probability. The encoder is updated for 60K steps, with a maximum of 16K tokens (*i.e.,* 640 seconds) per step, which takes 8–10 hours on 4 RTX A6000 GPUs. Visual corruption is applied to every sequence, randomly corrupting 10–50% of the sequence length with object occlusion, Gaussian noise, or blurring. Every clean audio sequence is corrupted by augmenting strong noise of babble, speech, music, or natural, at −10 dB SNR to 30–50% of the sequence length.

During fine-tuning, the encoder is frozen for the first 48K steps, while the decoder is trained using sequence-to-sequence negative log-likelihood as the AVSR loss function. The entire model is then trained for additional 12K steps. The visual corruption applied during fine-tuning is identical to that in the uptraining phase, while we randomly corrupt 25% of the audio signal by an SNR value sampled from a normal distribution with mean 0 and standard deviation 5. We note that all baseline models follow the same fine-tuning procedure as ours for a fair comparison: initialized from pretrained models and then fine-tuned with audio-visual corrupted inputs.

## 5.2 ROBUST AVSR BENCHMARK RESULTS

**LRS3 benchmark results with audio-visual joint corruption** Through our experiments, we address two questions: *(i) does representation learning with corrupted prediction task and multi-task learning approach improve the model's robustness to real-world audio-visual corruption?* and *(ii) does it guarantee generalizability to even unseen types of corruption?* The evaluation environments in this study are specifically challenging compared to previous works since there exists audio-visual joint corruption. In Table 1, we present the robust AVSR performance of our proposed model, CAV2vec, which is superior to baseline models in diverse conditions. AV-HuBERT (Shi et al., 2022b) and AV-data2vec (Lian et al., 2023) are based on a multimodal encoder and have been pretrained on

Table 1: Comparisons of WER (%) with our model and prior works on the LRS3 dataset (Afouras et al., 2018b). For audio corruption, babble, music, and natural noise are sampled from the MUSAN dataset (Snyder et al., 2015), while speech noise is sampled from the held-out set of LRS3. N-WER averages the results across all four audio noise types and five SNR (signal-to-noise ratio) values, while N ≥ S denotes noise-dominant scenarios, averaging over {-10, -5, 0} SNRs. For visual corruption, we evaluate on (a) object occlusion and noise, (b) occlusion by hands, and (c) pixelated face.

| | Method | Params | Babble, SNR (dB) = | | | | | | Speech, SNR (dB) = | | | | | | Music + Natural, SNR (dB) = | | | | | | N-WER | | Clean |
| | | | -10 | -5 | 0 | 5 | 10 | AVG | -10 | -5 | 0 | 5 | 10 | AVG | -10 | -5 | 0 | 5 | 10 | AVG | AVG | N≥S | ∞ |
|---|---|---|---|---|---|---|---|---|---|---|---|---|---|---|---|---|---|---|---|---|---|---|---|
| (a) Obj. + Noise | V-CAFE | 49M | 54.7 | 31.6 | 14.6 | 7.3 | 5.4 | 22.7 | 45.1 | 29.3 | 16.9 | 10.0 | 6.6 | 21.6 | 32.1 | 17.6 | 9.1 | 6.5 | 5.2 | 14.1 | 18.1 | 25.8 | 4.2 |
| | RAVEn | 673M | 43.5 | 25.4 | 8.5 | 4.0 | 2.8 | 16.9 | 58.9 | 42.3 | 17.9 | 5.1 | 3.1 | 25.5 | 23.3 | 11.3 | 5.4 | 3.3 | 2.5 | 9.2 | 15.2 | 23.0 | 2.3 |
| | BRAVEn | 673M | 41.1 | 22.2 | 6.1 | 2.5 | 1.8 | 14.7 | 45.6 | 30.4 | 14.9 | 5.2 | 2.2 | 19.7 | 20.3 | 8.3 | 3.5 | 2.1 | 1.8 | 7.2 | 12.2 | 18.7 | 1.8 |
| | AV-HuBERT | 325M | 30.6 | 14.4 | 5.1 | 2.7 | 2.1 | 11.0 | 7.7 | 4.3 | 3.0 | 2.2 | 2.0 | 3.9 | 10.9 | 5.2 | 3.0 | 2.3 | 1.8 | 4.6 | 6.0 | 8.6 | 1.6 |
| | AV-data2vec | 325M | 32.1 | 15.1 | 5.3 | 2.5 | 2.0 | 11.4 | 8.3 | 4.9 | 3.1 | 2.2 | 1.9 | 4.1 | 10.9 | 5.5 | 3.0 | 2.1 | 1.8 | 4.6 | 6.2 | 9.0 | 1.5 |
| | AV-RelScore | 437M | 30.1 | 14.3 | 5.3 | 2.5 | 1.7 | 10.8 | 6.8 | 4.4 | 2.8 | 2.3 | 2.0 | 3.7 | 10.5 | 5.3 | 2.8 | 2.1 | 1.8 | 4.5 | 5.9 | 8.4 | 1.6 |
| | **CAV2vec** | 325M | 25.8 | 11.7 | 4.4 | 2.4 | 1.8 | **9.2** | 5.9 | 3.6 | 2.5 | 2.1 | 1.8 | **3.2** | 9.6 | 4.3 | 2.6 | 1.8 | 1.7 | **4.0** | **5.1** | **7.2** | **1.5** |
| (b) Hands Occ. | V-CAFE | 49M | 57.2 | 31.9 | 13.7 | 7.5 | 5.3 | 23.1 | 45.7 | 29.3 | 17.1 | 10.0 | 6.8 | 21.8 | 32.3 | 18.0 | 9.2 | 6.2 | 5.4 | 14.2 | 18.3 | 26.2 | 4.1 |
| | RAVEn | 673M | 46.0 | 25.8 | 9.0 | 4.2 | 2.8 | 17.6 | 60.6 | 44.0 | 18.9 | 5.5 | 2.9 | 26.4 | 23.7 | 11.4 | 5.1 | 3.4 | 2.7 | 9.3 | 15.6 | 23.7 | 2.2 |
| | BRAVEn | 673M | 38.3 | 21.6 | 5.7 | 2.5 | 2.0 | 14.0 | 41.8 | 29.0 | 13.9 | 4.6 | 2.4 | 18.4 | 18.5 | 7.5 | 3.3 | 2.2 | 1.9 | 6.7 | 11.4 | 17.4 | 1.7 |
| | AV-HuBERT | 325M | 32.0 | 15.5 | 5.3 | 2.7 | 2.0 | 11.5 | 8.1 | 4.2 | 3.0 | 2.3 | 2.0 | 3.9 | 11.5 | 5.3 | 2.9 | 2.3 | 1.8 | 4.8 | 6.2 | 9.0 | 1.6 |
| | AV-data2vec | 325M | 33.2 | 15.6 | 5.7 | 2.6 | 2.0 | 11.8 | 8.1 | 4.7 | 2.8 | 2.3 | 1.9 | 4.0 | 12.3 | 6.0 | 2.9 | 2.2 | 1.9 | 5.0 | 6.5 | 9.4 | 1.5 |
| | AV-RelScore | 437M | 31.7 | 14.5 | 5.1 | 2.7 | 2.0 | 11.2 | 7.7 | 4.2 | 2.6 | 2.2 | 1.9 | 3.7 | 11.2 | 5.3 | 3.1 | 2.1 | 1.7 | 4.7 | 6.1 | 8.7 | 1.6 |
| | **CAV2vec** | 325M | 26.6 | 12.4 | 4.5 | 2.6 | 1.8 | **9.6** | 6.2 | 3.6 | 2.6 | 2.2 | 1.7 | **3.3** | 9.4 | 4.8 | 2.6 | 1.9 | 1.7 | **4.1** | **5.2** | **7.4** | **1.5** |
| (c) Pixelate | V-CAFE | 49M | 55.4 | 31.8 | 13.7 | 7.5 | 5.3 | 22.7 | 44.4 | 28.3 | 16.8 | 10.1 | 7.0 | 21.3 | 31.8 | 17.9 | 9.4 | 6.4 | 5.2 | 14.1 | 18.1 | 25.7 | 4.2 |
| | RAVEn | 673M | 43.7 | 25.2 | 8.5 | 3.6 | 2.8 | 16.8 | 60.0 | 42.8 | 18.2 | 5.0 | 3.2 | 25.8 | 23.4 | 10.9 | 5.2 | 3.2 | 2.6 | 9.1 | 15.2 | 23.1 | 2.3 |
| | BRAVEn | 673M | 39.1 | 21.6 | 5.7 | 2.7 | 1.9 | 14.2 | 42.4 | 31.0 | 13.3 | 4.8 | 2.2 | 18.7 | 18.5 | 7.6 | 3.4 | 2.2 | 1.9 | 6.7 | 11.6 | 17.7 | 1.7 |
| | AV-HuBERT | 325M | 29.8 | 13.6 | 5.0 | 2.6 | 1.9 | 10.6 | 7.4 | 4.1 | 2.7 | 2.1 | 2.0 | 3.7 | 10.9 | 5.1 | 2.8 | 2.2 | 1.9 | 4.6 | 5.8 | 8.3 | 1.6 |
| | AV-data2vec | 325M | 30.6 | 14.3 | 5.2 | 2.6 | 2.0 | 10.9 | 7.5 | 4.5 | 3.0 | 2.3 | 2.0 | 3.9 | 10.7 | 5.4 | 2.9 | 2.1 | 1.7 | 4.6 | 6.0 | 8.6 | 1.5 |
| | AV-RelScore | 437M | 30.1 | 13.1 | 5.2 | 2.6 | 2.0 | 10.6 | 7.3 | 4.3 | 3.0 | 2.2 | 1.9 | 3.7 | 10.3 | 5.0 | 3.1 | 2.1 | 1.8 | 4.5 | 5.8 | 8.3 | 1.5 |
| | **CAV2vec** | 325M | 26.0 | 12.0 | 4.7 | 2.5 | 1.9 | **9.4** | 5.8 | 3.6 | 2.5 | 2.3 | 1.7 | **3.2** | 9.6 | 4.2 | 2.6 | 1.9 | 1.7 | **4.0** | **5.1** | **7.3** | **1.5** |

Table 2: Performance comparison on the LRS3 dataset (Afouras et al., 2018b) with audio noise sampled from the DEMAND dataset (Thiemann et al., 2013). For each noisy environment, WER (%) is measured by randomly sampling the SNR value from the range [−10 dB, 10 dB].

| Method | (a) Object Occlusion + Noise | | | | | | | | | (b) Hands Occlusion | | | | | | | | | (c) Pixelated Face | | | | | | | | |
| | PARK | RIVER | CAFE | RESTO | CAFETER | METRO | STATION | MEETING | AVG | PARK | RIVER | CAFE | RESTO | CAFETER | METRO | STATION | MEETING | AVG | PARK | RIVER | CAFE | RESTO | CAFETER | METRO | STATION | MEETING | AVG |
|---|---|---|---|---|---|---|---|---|---|---|---|---|---|---|---|---|---|---|---|---|---|---|---|---|---|---|---|
| BRAVEn | 4.6 | 7.3 | 6.6 | 14.9 | 8.1 | 3.3 | 6.1 | 13.5 | 8.1 | 4.0 | 7.0 | 5.7 | 13.6 | 8.3 | 3.8 | 6.1 | 12.5 | 7.6 | 4.8 | 7.3 | 6.6 | 15.6 | 8.4 | 3.3 | 5.7 | 12.3 | 8.0 |
| AV-HuBERT | 3.4 | 4.6 | 5.1 | 10.2 | 5.9 | 2.7 | 4.1 | 3.9 | 5.0 | 3.6 | 5.1 | 5.3 | 11.2 | 6.7 | 2.7 | 4.2 | 4.5 | 5.4 | 3.4 | 4.7 | 4.6 | 10.4 | 6.1 | 2.8 | 3.8 | 3.8 | 4.9 |
| AV-data2vec | 3.4 | 4.5 | 5.1 | 10.3 | 6.2 | 2.7 | 4.1 | 4.4 | 5.1 | 3.3 | 5.0 | 5.8 | 11.9 | 6.5 | 3.2 | 4.1 | 4.4 | 5.5 | 3.4 | 4.7 | 5.0 | 9.3 | 5.5 | 3.0 | 4.1 | 3.9 | 4.9 |
| AV-RelScore | 3.4 | 4.5 | 5.1 | 9.3 | 5.4 | 2.8 | 3.9 | 3.8 | 4.8 | 3.0 | 5.2 | 5.1 | 10.8 | 6.2 | 2.8 | 3.8 | 4.6 | 5.2 | 3.2 | 4.9 | 4.6 | 10.0 | 6.1 | 2.7 | 3.7 | 3.9 | 4.9 |
| **CAV2vec** | 2.8 | 4.3 | 4.4 | 8.4 | 5.1 | 2.3 | 3.8 | 3.5 | **4.3** | 3.0 | 4.0 | 4.0 | 8.9 | 5.1 | 2.7 | 3.4 | 3.5 | **4.3** | 3.0 | 3.9 | 4.5 | 8.6 | 4.6 | 2.4 | 3.3 | 3.6 | **4.2** |

noise-augmented audio conditions, which result in outperforming RAVEn (Haliassos et al., 2023) and BRAVEn (Haliassos et al., 2024) that utilize two separate encoders for ASR and VSR. RAVEn and BRAVEn particularly suffer in severely corrupted environments, *i.e.,* SNR ≤ 0 dB. While AV-RelScore is specifically designed to address audio-visual joint corruption by assessing the modality reliability (Hong et al., 2023) and outperforms other baselines, it entails additional parameters for a scoring module within the encoder, making it hard to incorporate with pretrained models.

CAV2vec consistently demonstrates superior performance across all visual and audio corruption levels. It surpasses all baseline models under the object occlusion and visual noise condition, achieving an average N-WER of 5.1% (Table 1(a)), while AV-data2vec and AV-RelScore obtain 6.2% and 5.9%, respectively. Furthermore, CAV2vec shows effectiveness in generalizing to unseen types of corruption, with N-WER of 5.2% and 5.1% for (b) hands occlusion and (c) pixelated face, respectively, underscoring its practical applicability in real-world scenarios. Occlusion by hands poses a particularly challenging situation, as the obscured region is larger than that of the COCO objects (Lin et al., 2014), and there is visual similarity between hands and facial tones. While baseline models struggle with such unseen visual corruption type, showing increases in average N-WER (6.5% for AV-data2vec and 6.1% for AV-RelScore), CAV2vec maintains robustness, achieving an N-WER of 5.2%.

In the audio noise-dominant scenarios, characterized by an SNR value less than or equal to 0 dB (denoted as N ≥ S), it is important to fully leverage visual cues to compensate for impaired speech audio signal. In such conditions, corrupted video inputs can be particularly detrimental if the model is not robust to them. CAV2vec, with its unimodal corrupted prediction tasks, effectively learns cross-modal correlations under corruption, mitigating the recognition errors. We also highlight that our

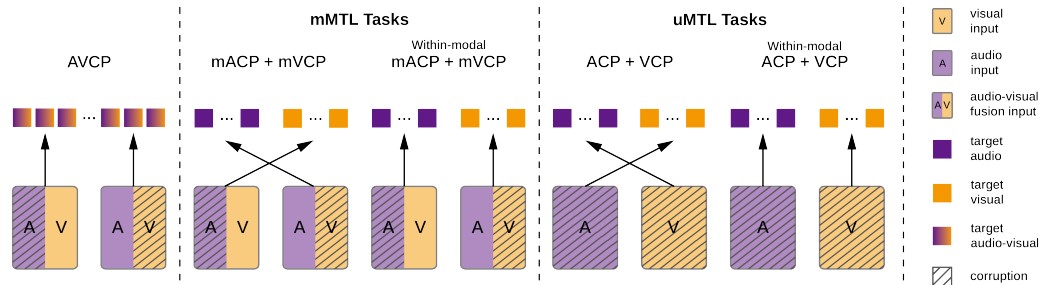

Figure 5: Our implemented strategies for corrupted prediction tasks. The AVCP task uses the audio-visual targets. For the multi-task learning (MTL) designs that utilize unimodal targets, mACP and mVCP tasks use multimodal inputs (mMTL), while ACP and VCP tasks use unimodal inputs (uMTL).

Table 4: Ablation study for the configuration of corrupted prediction tasks. We summarize the AVSR results (average N-WER) across visual corruption types (O: object occlusion + noise, H: hands occlusion, P: pixelate) and audio corruption types (MS: MUSAN and LRS3 noise, DM: DEMAND noise), same evaluation procedures as Tables 1 and 2. CRL: corrupted representation learning, mMTL: multimodal multi-task learning, uMTL: unimodal multi-task learning. Refer to Figure 5 for each setting. Note that the masked prediction task is utilized in every setup. Best and second best results.

| CRL | mMTL | uMTL | Tasks for corruption | O/MS | O/DM | H/MS | H/DM | P/MS | P/DM |
|---|---|---|---|---|---|---|---|---|---|
| ✗ | ✗ | ✗ | - | 6.2 | 5.1 | 6.5 | 5.5 | 6.0 | 4.9 |
| ✓ | ✗ | ✗ | AVCP | 5.3 | 4.5 | 5.6 | 4.7 | 5.6 | 4.8 |
| ✓ | ✓ | ✗ | mACP + mVCP | 5.1 | 4.2 | 5.4 | 4.5 | 5.3 | 4.3 |
| ✓ | ✓ | ✗ | mACP + mVCP (within-modal) | 5.2 | 4.4 | 5.4 | 4.6 | 5.3 | 4.5 |
| ✓ | ✓ | ✗ | mACP + mVCP + AVCP | 5.3 | 4.6 | 5.6 | 4.8 | 5.3 | 4.6 |
| ✓ | ✗ | ✓ | ACP + VCP | 5.1 | 4.3 | 5.2 | 4.3 | 5.1 | 4.2 |
| ✓ | ✗ | ✓ | ACP + VCP (within-modal) | 5.2 | 4.6 | 5.4 | 4.7 | 5.2 | 4.4 |
| ✓ | ✗ | ✓ | ACP + VCP + AVCP | 5.2 | 4.4 | 5.6 | 4.6 | 5.3 | 4.6 |
| ✓ | ✓ | ✓ | mACP + mVCP + ACP + VCP | 5.1 | 4.3 | 5.2 | 4.4 | 5.2 | 4.2 |

model consistently achieves state-of-the-art results even in the signal-dominant conditions, *i.e.,* high SNR values, demonstrating its versatility across varying levels of audio corruption. In addition, to validate the model's generalizability to real-world audio corruption, Table 2 presents the AVSR performance with DEMAND noise that includes more realistic environments. CAV2vec effectively achieves robust recognition capabilities in environments like indoor stores or outdoor public space.

**LRS2 benchmark results** The LRS2 dataset (Son Chung et al., 2017) consists of 224 hours of BBC video recordings, encompassing a wider variety of scenarios than LRS3, including news delivery, panel discussion, and indoor and outdoor interviews. This diversity allows for a more comprehensive evaluation of the model's generalizability. In Table 3, CAV2vec demonstrates strong performance on the corrupted LRS2 benchmark, further validating its robustness in real-world conditions. We note that all models compared are based on the LRS3-pretrained models, with LRS2 used only in the uptraining or fine-tuning phase.

Table 3: Comparisons of WER (%) with our model and prior works on the LRS2 dataset. We present N-WER, with babble (B), speech (S), music (M), and natural (N) noise types, as well as clean WER results. Visual corruption type is used as object occlusion + noise.

| Method | B | S | M | N | Clean |
|---|---|---|---|---|---|
| AV-HuBERT | 11.6 | 5.3 | 6.1 | 6.0 | 3.0 |
| AV-data2vec | 11.5 | 5.6 | 6.5 | 6.2 | 3.0 |
| AV-RelScore | 11.1 | 4.8 | 5.9 | 5.5 | 2.9 |
| **CAV2vec** | **8.9** | **4.4** | **5.1** | **4.9** | **2.7** |

## 5.3 ANALYSIS

### 5.3.1 ABLATION STUDY FOR CORRUPTED PREDICTION TASKS

In designing our corrupted prediction tasks, we have introduced the AVCP task in Eq. (3), along with the unimodal ACP and VCP tasks in Eq. (4). Additionally, we explore the mACP and mVCP tasks for a more comprehensive analysis. Figure 5 illustrates and compares these tasks, as well as

within-modal task designs. The within-modal ACP and VCP tasks are implemented to align the corrupted inputs and targets within the same modality. Table 4 summarizes the performance results for each task design.

The first observation is that without incorporating the suggested corrupted representation learning, the model struggles to achieve robust speech recognition. Therefore, it is crucial to employ any forms of corrupted prediction task. When utilizing multimodal inputs, the mACP + mVCP tasks consistently outperform the AVCP task alone, and even the combination with AVCP. This indicates that distilling knowledge through unimodal targets for the corrupted modality shows effectiveness. Besides, using unimodal inputs (ACP + VCP) proves even more effective, as these tasks generally outperform the multimodal input tasks (mACP + mVCP). We also note that within-modal strategies are less effective than cross-modal approaches, as they do not directly target audio-visual correlations.

These findings align with our hypothesis, as depicted in Figure 3, that directly addressing the cross-modal alignment improves the correlation between the two modalities, which is crucial in generating robust audio-visual features. Since the masked prediction task is maintained for learning contextualized multimodal representations, the AVCP or mACP + mVCP tasks become redundant when unimodal tasks are being employed. We thus separate the multi-task strategy as leveraging only unimodal inputs for corrupted prediction and multimodal inputs for masked prediction.

### 5.3.2 SENSITIVITY STUDY ON CORRUPTION RATIOS

We evaluate the model across varying corruption ratios for both audio ($p_{\text{corrupt}}^a$) and visual ($p_{\text{corrupt}}^v$) sequences to examine the impact of amount of corruption in learning robustness. As shown in Table 5, higher corruption ratios naturally reduce the empirical proportion of clean inputs ($\hat{p}_{\text{clean}}$). The result reveals a trade-off between the performance in highly noisy environments (N ≥ S) and less corrupted settings (N < S and Clean), depending on the model's exposure to clean or corrupted sequences during training. While higher corruption ratios significantly improve performance in noisy conditions, it is also essential for the AVSR model to maintain reliability in less corrupted scenarios. To balance this trade-off,

Table 5: Sensitivity study on the corruption ratios in CAV2vec. Object occlusion with Gaussian noise and blurring is used for visual corruption, along with the MUSAN audio corruption, as in Table 1(a).

| | | | | N-WER | | | |
|---|---|---|---|---|---|---|---|
| $p_{\text{corrupt}}^v$ | $p_{\text{corrupt}}^a$ | $\hat{p}_{\text{clean}}$ | AVG | N ≥ S | N < S | Clean |
| 0.1–0.5 | 0.3–0.5 | 0.15 | 5.1 | 7.2 | **1.9** | **1.5** |
| 0.1 | 0.3 | 0.22 | 5.3 | 7.4 | 2.0 | 1.6 |
| 0.3 | 0.5 | 0.14 | 5.0 | 7.1 | 2.0 | **1.5** |
| 0.5 | 0.7 | 0.09 | **4.7** | **6.5** | 2.0 | 1.6 |
| 0.7 | 0.9 | 0.04 | **4.7** | **6.5** | 2.1 | 1.6 |

we randomly sample the visual corruption ratio from 0.1–0.5 and audio corruption ratio from 0.3–0.5, ensuring a balance between clean and corrupted inputs. In terms of masking frames, we follow Shi et al. (2022b); Zhang et al. (2023) by using a 0.3 masking ratio for video and 0.8 for audio. For both corruption and masking, higher ratios for audio is necessary as audio holds more critical information in speech. However, we note that masking is applied after corruption, and we do not allow overlap between masked and corrupted frames, which results in effective masking ratios of roughly 0.1 for video and 0.2 for audio. We empirically observed that adjusting the masking ratios has little effect on the CAV2vec's performance.

## 6 CONCLUSION

In this paper, we presented CAV2vec, a novel audio-visual representation learning framework designed to address the challenges of audio-visual joint corruption in speech recognition. By employing a corrupted prediction task, CAV2vec enhances multimodal robustness, while incorporating a unimodal multi-task learning strategy to improve cross-modal alignment shows effectiveness. Our experiments on robust AVSR benchmarks including LRS3 and LRS2 demonstrate that CAV2vec significantly outperforms existing baseline models, consistently exhibiting superior performance across a variety of corrupted environments. Particularly in challenging audio noise-dominant scenarios, CAV2vec effectively aligns corrupted modalities, leading to more reliable and robust audio-visual fusion. Additionally, our model showcases strong generalization abilities, achieving state-of-the-art results even in unseen corruption types such as pixelated faces with public audio noise. These findings establish CAV2vec as a robust, adaptable framework for handling corrupted audio-visual data, setting a new benchmark for multimodal speech recognition systems.

## ETHICS STATEMENT

Our research focuses on advancing representation learning methods for audio-visual speech data, specifically addressing the challenge of multimodal corruption in speech recognition systems. This study does not involve human subjects, nor does it present any potential ethical concerns related to harmful insights, conflicts of interest, discrimination, or bias. The datasets used in this research, including the newly introduced corruption types (*e.g.,* hands occlusion or background noise recordings), are publicly available and widely used in the speech recognition community, ensuring our compliance with data privacy and legal standards.

## REPRODUCIBILITY STATEMENT

To ensure the reproducibility of our work, we provide a comprehensive description of our model architecture and training process in Section 5.1. Additional details, including hyperparameters, training strategies, and corruption techniques for CAV2vec, can be found in Appendix A.2 and A.3. Appendix A.1 outlines the details of each baseline model used in the experiments, including the implementation procedures to fine-tune the models on the corrupted audio-visual data. We plan to release the model checkpoint of CAV2vec to facilitate future research.

## ACKNOWLEDGMENTS

This work was supported by Institute of Information & communications Technology Planning & Evaluation (IITP) grant funded by the Korea government (MSIT) [No.2022-0-00641, XVoice: Multi-Modal Voice Meta Learning, 90%] and [No.2019-0-00075, Artificial Intelligence Graduate School Program (KAIST), 10%].

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

# APPENDIX

## A  IMPLEMENTATION DETAILS

### A.1  IMPLEMENTATION OF BASELINE MODELS

We present the baseline models used in our experiments and provide the details on how they have been (re-)implemented.

**V-CAFE** (Hong et al., 2022)  is an end-to-end supervised model for AVSR, designed to capture the lip movement and generate a noise reduction mask by utilizing visual context. The model is built on a Conformer-Transformer encoder-decoder architecture and employs a joint CTC/Attention loss function (Kim et al., 2017). V-CAFE incorporates visual context through cross-modal attention to generate a noise reduction mask, where the generated mask is applied to the encoded audio features to mitigate noisy audio representations.

For our experiments, we use the pretrained encoder from the publicly available V-CAFE checkpoint[1], which has been trained on the LRS3 dataset. To ensure a consistent experimental setup across all baselines and our model, we fine-tune the V-CAFE model by initializing the Transformer decoder and training it for 120,000 steps with a learning rate of $2 \times 10^{-3}$. The encoder remains frozen for the first 48,000 steps, after which the entire model is updated over the remaining 72,000 steps.

**RAVEn and BRAVEn** (Haliassos et al., 2023; 2024)  are self-supervised learning ASR and VSR models that encode masked inputs and predict contextualized targets generated by momentum encoder teachers. In both models, two unimodal encoders are jointly trained, each serving as a teacher for the cross-modal student encoder. Specifically, the audio student predicts outputs from both audio and video teachers, while the video student predicts only audio targets. BRAVEn is the upgraded version of RAVEn, slightly modifying the self-distillation framework with different hyperparameters to more emphasize ASR than VSR.

We utilize the pretrained encoders of RAVEn and BRAVEn from the public repository[2]. Both ASR and VSR encoders are loaded, and these models are used to encode the normalized raw audio waveform and video frames, respectively. Although RAVEn and BRAVEn were not originally designed as multimodal models, we follow the approach of Haliassos et al. (2024) to implement an AVSR framework by fusing the encoded features from each modality. The fusion audio-visual features are then fed into an initialized Transformer decoder. Thus, the modality-fusion MLP layer is also trained with the decoder. We train the model for 120,000 steps with a learning rate of $2 \times 10^{-3}$ while the encoder is frozen for the first 96,000 steps. Due to the absence of a pretrained multimodal encoder and the fact that these models were not trained on noise-augmented data, their AVSR performance in corrupted environments is suboptimal despite of their large number of model parameters.

The RAVEn and BRAVEn results in Table 1 exhibit poor performances than those reported in the original paper. This discrepancy can be attributed to different evaluation settings. Our settings are designed to assess noise-robust audio-visual models under real-world conditions, where the type and extent of modality corruption are unpredictable. These results (WER: RAVEn 2.3% and BRAVEn 1.8%) are measured under visual corruption, whereas the published results pertain to clean audio conditions using a standalone ASR model. Although standalone ASR models excel in clean audio environments, their performance significantly degrades under noisy conditions.

BRAVEn has reported low-resource AVSR performance (WER: RAVEn 4.7% and BRAVEn 4.0%), but it does not provide results in high-resource settings or under audio-visual corruption. Moreover, the specific hyperparameters used for fine-tuning in conjunction with pretrained ASR and VSR encoders and a decoder have not been detailed. The performance gap might also stem from the larger unlabeled dataset and self-training technique during pretraining and the use of a language model during inference, which were not employed in our experimental setup. Also, while (B)RAVEn used CTC/attention loss for fine-tuning, we only used attention loss to ensure a fair comparison across all

---

[1] https://github.com/ms-dot-k/AVSR
[2] https://github.com/ahaliassos/raven

models. These variations in decoding approaches can influence outcomes, although they could be orthogonally applied to any models.

**AV-HuBERT and AV-data2vec** (Shi et al., 2022a; Lian et al., 2023) are self-supervised learning models for audio-visual multimodal processing within a single framework, both using masked inputs to predict unmasked targets. They share the same structure for the encoder, with 24 Transformer blocks. The key difference lies in how they generate the targets: AV-HuBERT uses the cluster indices of MFCC (mel-frequency cepstral coefficient) features, while AV-data2vec uses the EMA teacher's output sequence. AV-HuBERT updates its targets after each iteration of training using the current model, whereas AV-data2vec generates online targets with a self-evolving teacher. Both methods are effective in learning contextualized audio-visual multimodal features.

Since AV-HuBERT pretrained models are publicly available[3] but AV-data2vec models are not, we implement AV-data2vec on top of the AV-HuBERT pretrained model, and uptrained it for 60,000 steps in a similar manner to CAV2vec. We use the Adam optimizer with a weight decay of 0.01 and a learning rate of $2 \times 10^{-4}$. We fine-tune the decoder for both AV-HuBERT and AV-data2vec models, building on the respective pretrained encoders. The fine-tuning process employs an attention-based sequence-to-sequence cross-entropy loss, with accuracy as the validation metric. The initial learning rate is $10^{-3}$, scheduled with 20,000 warmup steps followed by decaying over the next 40,000 steps. During fine-tuning, the encoder remains frozen for the first 48,000 steps, updating only the decoder. After 48,000 steps, both the encoder and decoder are updated for the final 12,000 steps.

**AV-RelScore** (Hong et al., 2023) leverages a reliability scoring module (RelScore) to assess the reliability of each input modality at every frame. RelScore modules are appended after the feature extractor for each modality and consist of three convolutional layers followed by a sigmoid activation, which outputs a scalar value for each frame. The original model is based on the V-CAFE backbone (Hong et al., 2022), which has a smaller architecture and subpar performance on large-scale datasets. To ensure a fair comparison with our baselines, we implement RelScore on the AV-HuBERT-LARGE backbone, leveraging its pretrained knowledge. For training AV-RelScore, we freeze the encoder and feature extractors, training only the RelScore modules and the decoder for 50,000 steps. Afterward, we update the entire model, including the encoder, for an additional 40,000 steps. Since the RelScore modules introduce additional parameters and are located before the pretrained encoder, AV-RelScore requires more fine-tuning steps until convergence than required by AV-HuBERT.

### A.2    TRAINING DETAILS OF CAV2VEC

We perform uptraining using a Transformer-based AV-HuBERT-LARGE architecture via unimodal corrupted prediction tasks. Both audio and video data are processed at 25 frames per second (fps) and augmented with various corruptions (refer to Figure 2). Each sample sequence is trimmed to a maximum length of 400 frames during pre-processing. For audio, 25% of the sequences are applied with noise at SNR = 0 dB, following the noise augmentation strategy from Shi et al. (2022b), while the remaining 75% undergo partial corruption at SNR = −10 dB. A single chunk within each sequence is corrupted, with the corruption length randomly selected between 30–50% of the sequence length. The visual modality is corrupted at a frequency of 1, where frequency denotes the number of visual corruption events in the entire sequence. Object occlusion applied to the speaker's lips occurs once, followed by Gaussian noise or blurring, each with a probability of 0.3. The visual corruption length is randomly selected as 10–50%.

We also apply frame masking for contextualized representation learning. Following the strategy in previous works (Shi et al., 2022a), 80% of audio frames are masked, with each mask segment lasting 10 frames, while 30% of video frames are masked, with each segment lasting 5 frames. The high audio masking ratio helps the model focus on the most relevant information from the audio context. However, we note that masking is applied after corruption, and we do not allow overlap between masked and corrupted frames. Therefore, the effective masking ratio is lower than the initially set probability.

Additionally, modality dropout is applied to both audio and visual inputs, each with a dropout rate of 0.25. In our implementation of CAV2vec with unimodal multi-task learning, we use audio targets

---

[3]https://github.com/facebookresearch/av_hubert

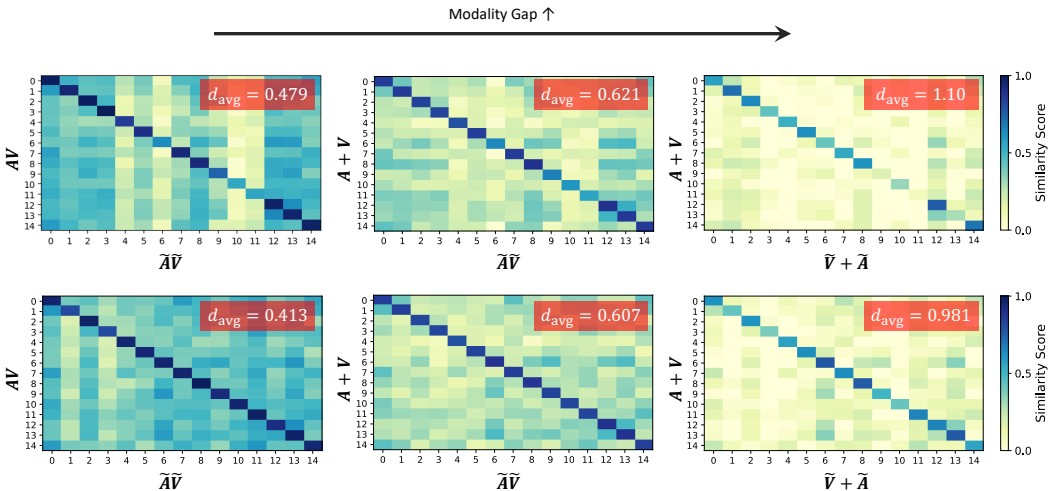

Figure 6: Visualization of the modality gap of (1st row) AV-data2vec and (2nd row) CAV2vec. Each column shows representation comparisons between different modality configurations: (1st column) multimodal-to-multimodal comparisons, (2nd column) unimodal-to-multimodal comparisons (averaged across audio-to-multimodal and video-to-multimodal), and (3rd column) unimodal-to-unimodal comparisons in a cross-modal manner. $d_{\text{avg}}$ represents the distance between the average embeddings of each modality.

when the audio input is dropped out (video-only input) and video targets when the video input is dropped out (audio-only input). The audio-visual target is always used for the masked prediction task. CAV2vec is optimized using a multi-task loss function that includes ACP + VCP losses, each weighted at 1.0 ($\lambda_{\text{ACP}} = \lambda_{\text{VCP}} = 1.0$). The masked prediction loss coefficient is weighted at 1.0, while the MLM loss is weighted at 2.0 ($\lambda_{\text{MASK}} = 1.0$, $\lambda_{\text{MLM}} = 2.0$), balancing the scales of self-distillation regression loss and MLM cross-entropy loss.

We use the Adam optimizer (Kingma, 2014) with a weight decay of 0.01 and a learning rate of $10^{-4}$. The EMA decaying parameter $\eta$ starts from 0.99 and increases up to 0.999 over training. The model is updated for 60,000 steps, using a polynomial decay learning rate scheduler with a warmup phase of the first 5,000 updates. Each model update consumes a batch of 16,000 tokens, equivalent to 640 seconds of audio-visual data. For fine-tuning, we largely follow AV-HuBERT and apply the same procedure to all baseline models. This involves initializing the 9-layer Transformer decoder and training the AVSR task with a sequence-to-sequence negative log-likelihood loss. For simplicity of fine-tuning, we do not use connectionist temporal classification (CTC) loss (Graves et al., 2006).

## A.3 EVALUATION DETAILS

For evaluation, we assess the model's performance under various audio-visual joint corruption scenarios, including corruption types that were not encountered during training. For visual corruption types, object occlusion and Gaussian noise (or blurring) are applied with a frequency of 1 each, consistent with the training phase. For unseen visual corruptions, *i.e.,* hands occlusion and pixelated face, we randomly sample a frequency from $\{1, 2, 3\}$, resulting in an average of two corruption occurrences per sequence, similar to the object occlusion + noise. The model's AVSR performance is measured using the word error rate (WER) across five SNR values: $\{-10, -5, 0, 5, 10\}$. Audio corruption is introduced using noise from the MUSAN dataset, including babble, music, and natural noise, as well as LRS3 speech noise. To ensure that there is no speaker overlap between the training and test sets, LRS3 speech noise is generated using distinct speakers. Additionally, when evaluating the model with unseen DEMAND noise, we corrupt audio by randomly sampling an SNR value from the range $[-10, 10]$ for each environment category.

## B VISUALIZATION OF MODALITY GAP

Figure 6 visualizes the modality gap across different comparisons: multimodal-to-multimodal, unimodal-to-multimodal, and unimodal-to-unimodal. Similar to Figure 4, we measure the repre-

sentation similarity scores between sequence features, along with the distance between the average embeddings over 50,000 samples. The diagonal line in the similarity matrix effectively captures the modality gap, as it indicates the distance between different modalities for the same sample. The results demonstrate that the modality gap widens as more unimodal features are introduced, following the intuition that the unimodal-to-unimodal prediction task best improves cross-modal alignment. According to Table 4, representation learning is enhanced when directly targeting the larger modality gap (③ > ② > ① in Figure 3). In addition, we present the representation similarity of CAV2vec in the second row of Figure 6, observing that the modality gaps are smaller compared to AV-data2vec.

## C  ADDITIONAL EXPERIMENTS

### C.1  ABLATION STUDY FOR CORRUPTED PREDICTION TASKS

In Table 4, we have investigated different configurations of the proposed corrupted prediction tasks. To clearly outline the definitions for each acronym used, these are summarized in Table 6.

Table 6: Summary of notations for each task or loss in the CAV2vec framework.

| Notation | Description | Input modality | Target modality |
|---|---|---|---|
| MP | masked prediction | - | - |
| CP | corrupted prediction | - | - |
| AVCP | audio-visual CP | AV | AV |
| mACP | multimodal audio CP | AV | A |
| mVCP | multimodal visual CP | AV | V |
| ACP | unimodal audio CP | V | A |
| VCP | unimodal visual CP | A | V |

Table 7: Ablation study for the configuration of corrupted prediction tasks. We summarize the AVSR results (average N-WER) across visual corruption types (O: object occlusion + noise, H: hands occlusion, P: pixelate) and audio corruption types (MS: MUSAN and LRS3 noise, DM: DEMAND noise), same evaluation procedures as Tables 1 and 2. CRL: corrupted representation learning, mMTL: multimodal multi-task learning, uMTL: unimodal multi-task learning. Refer to Figure 5 for each setting. The masked prediction task is utilized in every setup. (w) denotes the within-modal strategies.

| CRL | mMTL | uMTL | Tasks for corruption | O/MS | O/DM | H/MS | H/DM | P/MS | P/DM |
|---|---|---|---|---|---|---|---|---|---|
| ✗ | ✗ | ✗ | - | 6.2 | 5.1 | 6.5 | 5.5 | 6.0 | 4.9 |
| ✓ | ✗ | ✗ | AVCP | 5.3 | 4.5 | 5.6 | 4.7 | 5.6 | 4.8 |
| ✓ | ✓ | ✗ | mACP + mVCP | **5.1** | **4.2** | 5.4 | 4.5 | 5.3 | 4.3 |
| ✓ | ✓ | ✗ | mACP (w) + mVCP (w) | 5.2 | 4.4 | 5.4 | 4.6 | 5.3 | 4.5 |
| ✓ | ✓ | ✗ | mACP + mVCP + AVCP | 5.3 | 4.6 | 5.6 | 4.8 | 5.3 | 4.6 |
| ✓ | ✗ | ✓ | ACP + VCP | **5.1** | 4.3 | **5.2** | **4.3** | **5.1** | **4.2** |
| ✓ | ✗ | ✓ | ACP (w) + VCP (w) | 5.2 | 4.6 | 5.4 | 4.7 | 5.2 | 4.4 |
| ✓ | ✗ | ✓ | ACP + VCP + ACP (w) | **5.1** | 4.4 | 5.4 | 4.5 | 5.2 | 4.3 |
| ✓ | ✗ | ✓ | ACP + VCP + VCP (w) | **5.1** | 4.3 | 5.3 | 4.5 | 5.3 | 4.5 |
| ✓ | ✗ | ✓ | ACP + VCP + ACP (w) + VCP (w) | **5.1** | 4.5 | 5.3 | 4.7 | 5.2 | 4.3 |
| ✓ | ✗ | ✓ | ACP + VCP + AVCP | 5.2 | 4.4 | 5.6 | 4.6 | 5.3 | 4.6 |
| ✓ | ✓ | ✓ | mACP + mVCP + ACP + VCP | **5.1** | 4.3 | **5.2** | 4.4 | 5.2 | **4.2** |
| ✓ | ✓ | ✓ | mACP (w) + mVCP (w) + ACP (w) + VCP (w) | 5.2 | 4.3 | 5.4 | 4.6 | 5.3 | 4.4 |

In addition to the results presented in Table 4, we provide complete results for the ablation study on corrupted prediction tasks in Table 7. We have included additional results on within-modal strategies, as Haliassos et al. (2023) explored similar training strategies for unimodal encoders. They concluded that, while the VSR student model benefits from using an ASR teacher, the ASR student model requires guidance from both ASR and VSR teachers. This is because the ASR model provides more contextual information, leading to effective training of both encoders.

Table 8: Performance comparison on the LRS3 dataset (Afouras et al., 2018b) with audio noise sampled from the DEMAND dataset (Thiemann et al., 2013). For each noisy environment, WER (%) is measured by randomly sampling the SNR value from the range [−10 dB, 10 dB].

| Method | PARK | RIVER | CAFE | RESTO | CAFETERIA | METRO | STATION | MEETING | KITCHEN | LIVING | WASH | FIELD | HALL | OFFICE | SQUARE | TRAFFIC | BUS | CAR | AVG |
|---|---|---|---|---|---|---|---|---|---|---|---|---|---|---|---|---|---|---|---|
| BRAVEn | 4.0 | 7.0 | 5.7 | 13.6 | 8.3 | 3.8 | 6.1 | 12.5 | 2.2 | 3.9 | 1.7 | 1.9 | 2.2 | 1.8 | 2.9 | 3.1 | 2.0 | 1.8 | 4.7 |
| AV-HuBERT | 3.6 | 5.1 | 5.3 | 11.2 | 6.7 | **2.7** | 4.2 | 4.5 | 2.2 | 3.2 | 1.6 | **1.8** | **1.9** | 1.8 | 2.6 | **2.5** | 2.0 | 1.6 | 3.6 |
| AV-data2vec | 3.3 | 5.0 | 5.8 | 11.9 | 6.5 | 3.2 | 4.1 | 4.4 | 2.2 | 3.2 | 1.6 | **1.8** | 2.2 | 1.8 | 2.5 | 2.7 | **1.9** | **1.5** | 3.6 |
| AV-RelScore | **3.0** | 5.2 | 5.1 | 10.8 | 6.2 | 2.8 | 3.8 | 4.6 | 2.2 | 3.2 | 1.7 | 1.9 | 2.1 | 1.7 | 2.2 | 3.0 | 2.1 | 1.7 | 3.5 |
| **CAV2vec** | **3.0** | **4.0** | **4.0** | **8.9** | **5.1** | **2.7** | **3.4** | **3.5** | **1.8** | **2.8** | **1.5** | 1.9 | **1.9** | **1.6** | **2.1** | 2.6 | **1.9** | 1.6 | **3.0** |

(a) Hands Occlusion

| Method | PARK | RIVER | CAFE | RESTO | CAFETERIA | METRO | STATION | MEETING | KITCHEN | LIVING | WASH | FIELD | HALL | OFFICE | SQUARE | TRAFFIC | BUS | CAR | AVG |
|---|---|---|---|---|---|---|---|---|---|---|---|---|---|---|---|---|---|---|---|
| BRAVEn | 4.8 | 7.3 | 6.6 | 15.6 | 8.4 | 3.3 | 5.7 | 12.3 | 2.5 | 3.9 | **1.7** | **1.7** | 2.2 | 1.8 | 2.8 | 3.2 | 2.1 | 1.7 | 4.9 |
| AV-HuBERT | 3.4 | 4.7 | 4.6 | 10.4 | 6.1 | 2.8 | 3.8 | 3.8 | 2.1 | 3.0 | **1.7** | **1.7** | 1.9 | 1.7 | 2.5 | 2.4 | 1.8 | 1.6 | 3.3 |
| AV-data2vec | 3.4 | 4.7 | 5.0 | 9.3 | 5.5 | 3.0 | 4.1 | 3.9 | 2.1 | 3.2 | **1.7** | 1.8 | 2.1 | **1.6** | 2.3 | 2.6 | 1.8 | **1.5** | 3.3 |
| AV-RelScore | 3.2 | 4.9 | 4.6 | 10.0 | 6.1 | 2.7 | 3.7 | 3.9 | **1.9** | 3.0 | 1.8 | 1.8 | 2.0 | 1.8 | 2.4 | **2.3** | 1.8 | 1.7 | 3.3 |
| **CAV2vec** | **3.0** | **3.9** | **4.5** | **8.6** | **4.6** | **2.4** | **3.3** | **3.6** | **1.9** | **2.6** | **1.7** | **1.7** | **1.8** | 1.7 | **2.2** | 2.4 | **1.7** | **1.5** | **2.9** |

(b) Pixelated Face

Table 9: Comparisons of WER (%) with our model and prior works on the LRS2 dataset (Son Chung et al., 2017). We follow the experimental setup as in Table 1.

| | Method | Params | Babble, SNR (dB) = | | | | | | Speech, SNR (dB) = | | | | | | Music + Natural, SNR (dB) = | | | | | | N-WER | | Clean |
|---|---|---|---|---|---|---|---|---|---|---|---|---|---|---|---|---|---|---|---|---|---|---|---|
| | | | -10 | -5 | 0 | 5 | 10 | **AVG** | -10 | -5 | 0 | 5 | 10 | **AVG** | -10 | -5 | 0 | 5 | 10 | **AVG** | **AVG** | N≥S | ∞ |
| **(a) Object** | AV-HuBERT | 325M | 28.8 | 15.2 | 6.7 | 3.9 | 3.5 | 11.6 | 9.4 | 5.6 | 4.3 | 3.6 | 3.4 | 5.3 | 11.7 | 6.8 | 4.7 | 3.7 | 3.3 | 6.0 | 7.2 | 9.7 | 3.0 |
| | AV-data2vec | 325M | 28.1 | 14.8 | 6.8 | 4.3 | 3.7 | 11.5 | 9.6 | 6.2 | 5.0 | 4.0 | 3.5 | 5.6 | 12.3 | 7.2 | 5.1 | 3.8 | 3.4 | 6.4 | 7.5 | 10.0 | 3.0 |
| | AV-RelScore | 437M | 28.5 | 14.3 | 5.8 | 3.5 | 3.2 | 11.1 | 8.3 | 5.4 | 4.1 | 3.3 | 2.9 | 4.8 | 11.9 | 6.3 | 4.1 | 3.3 | 2.9 | 5.7 | 6.8 | 9.2 | 2.9 |
| | **CAV2vec** | 325M | 21.2 | 10.9 | 5.5 | 3.7 | 3.1 | **8.9** | 7.1 | 4.9 | 3.6 | 3.2 | 3.2 | **4.4** | 9.4 | 5.5 | 3.9 | 3.3 | 3.0 | **5.0** | **5.8** | **7.5** | **2.7** |
| **(b) Hands** | AV-HuBERT | 325M | 28.9 | 15.0 | 7.0 | 4.1 | 3.7 | 11.7 | 8.6 | 5.7 | 4.2 | 3.7 | 3.4 | 5.1 | 12.7 | 7.1 | 4.5 | 3.6 | 3.2 | 6.2 | 7.3 | 9.8 | 3.0 |
| | AV-data2vec | 325M | 30.2 | 15.2 | 7.4 | 4.3 | 3.4 | 12.1 | 9.7 | 5.9 | 4.6 | 3.8 | 3.7 | 5.5 | 13.3 | 7.2 | 4.7 | 3.8 | 3.5 | 6.5 | 7.6 | 10.3 | 3.2 |
| | AV-RelScore | 437M | 34.3 | 15.7 | 6.4 | 3.7 | 3.1 | 12.6 | 10.0 | 6.2 | 4.0 | 3.5 | 3.0 | 5.3 | 14.2 | 7.2 | 4.5 | 3.4 | 3.0 | 6.5 | 7.7 | 10.7 | **2.7** |
| | **CAV2vec** | 325M | 21.2 | 12.2 | 6.4 | 3.7 | 3.2 | **9.3** | 7.6 | 5.8 | 4.0 | 3.5 | 3.2 | **4.8** | 10.4 | 6.3 | 4.5 | 3.6 | 3.5 | **5.7** | **6.4** | **8.3** | **2.7** |
| **(c) Pixelate** | AV-HuBERT | 325M | 28.3 | 14.7 | 6.1 | 4.1 | 3.5 | 11.4 | 9.2 | 5.8 | 4.0 | 3.8 | 3.3 | 5.2 | 12.4 | 7.4 | 4.6 | 3.6 | 3.2 | 6.2 | 7.3 | 9.7 | 2.9 |
| | AV-data2vec | 325M | 28.9 | 15.3 | 6.9 | 4.6 | 3.6 | 11.9 | 9.6 | 6.4 | 4.9 | 4.0 | 3.4 | 5.6 | 12.1 | 6.7 | 4.8 | 3.8 | 3.4 | 6.1 | 7.4 | 9.9 | 3.1 |
| | AV-RelScore | 437M | 34.1 | 16.0 | 6.1 | 4.0 | 3.2 | 12.7 | 9.9 | 5.6 | 4.2 | 3.4 | 3.1 | 5.3 | 14.0 | 6.8 | 4.3 | 3.3 | 3.0 | 6.3 | 7.6 | 10.5 | **2.7** |
| | **CAV2vec** | 325M | 22.2 | 12.5 | 6.2 | 4.3 | 3.3 | **9.7** | 7.9 | 5.6 | 4.4 | 3.5 | 3.2 | **4.9** | 10.0 | 6.6 | 4.3 | 3.5 | 3.4 | **5.6** | **6.4** | **8.4** | **2.7** |

We can similarly incorporate ACP and VCP tasks in a within-modal manner, which allows us to experiment with an audio-to-audio guidance approach. However, in Table 7, adding the within-modal ACP task has not gained improvement, nor has the within-modal VCP task. We find that only using cross-modal ACP + VCP tasks is sufficient, even surpassing the task configurations with mACP + mVCP tasks. This is distinct from Haliassos et al. (2023) where within-modal strategy is crucial for ASR, while our model is a unified multimodal framework that requires cross-modal knowledge.

## C.2 ADDITIONAL DEMAND NOISE TYPES

The original DEMAND dataset (Thiemann et al., 2013) contains 18 categories of recorded environments. In Table 2, we have presented results for 8 out of these categories, specifically excluding relatively quieter environments. In Table 8, we provide full results across all 18 categories, which include the following noise environments: park, river, cafe, restaurant, cafeteria, metro (subway), public station, meeting room, kitchen, living room, washroom, sports field, hallway, office, public square, traffic intersection, bus, and car. Still, CAV2vec generally outperforms the baselines in the remaining, relatively less noisy environments.

## C.3 FULL RESULTS OF LRS2 EVALUATION

In Tables 9 and 10, we show the full results of our evaluation on the LRS2 benchmark (Son Chung et al., 2017), with the same experimental setup as LRS3 evaluation. CAV2vec outperforms the baselines across all audio-visual corruption, effectively demonstrating the model's robustness under real-world conditions.

Table 10: Performance comparison on the LRS2 dataset (Son Chung et al., 2017) with audio noise sampled from the DEMAND dataset (Thiemann et al., 2013). We follow the experimental setup as in Table 2.

| | (a) Object Occlusion + Noise | | | | | | | | | (b) Hands Occlusion | | | | | | | | | (c) Pixelated Face | | | | | | | | |
|---|---|---|---|---|---|---|---|---|---|---|---|---|---|---|---|---|---|---|---|---|---|---|---|---|---|---|---|
| Method | PARK | RIVER | CAFE | RESTO | CAFETER | METRO | STATION | MEETING | **AVG** | PARK | RIVER | CAFE | RESTO | CAFETER | METRO | STATION | MEETING | **AVG** | PARK | RIVER | CAFE | RESTO | CAFETER | METRO | STATION | MEETING | **AVG** |
| AV-HuBERT | 4.8 | 5.2 | 5.8 | 11.6 | 7.1 | 4.2 | 5.5 | 6.0 | 6.3 | 5.1 | 5.7 | 6.5 | 12.5 | 7.6 | 4.2 | 5.4 | 6.2 | 6.6 | 4.6 | 6.5 | 6.4 | 10.1 | 7.4 | 4.1 | 5.0 | 6.3 | 6.3 |
| AV-data2vec | 5.0 | 6.6 | 7.0 | 11.7 | 6.9 | 4.5 | 5.4 | 6.2 | 6.7 | 5.2 | 6.0 | 7.7 | 12.4 | 7.6 | 4.4 | 5.9 | 6.3 | 6.9 | 5.1 | 5.7 | 6.9 | 10.9 | 7.4 | 4.2 | 5.5 | 6.1 | 6.5 |
| AV-RelScore | 4.7 | 5.9 | 6.1 | 12.2 | 7.5 | 3.8 | 5.0 | 6.5 | 6.5 | 5.0 | 5.9 | 6.7 | 12.9 | 8.8 | 4.6 | 5.0 | 6.5 | 6.9 | 5.1 | 5.8 | 6.7 | 11.8 | 7.5 | 4.0 | 5.2 | 6.4 | 6.6 |
| **CAV2vec** | 5.1 | 5.7 | 6.1 | 9.4 | 6.8 | 4.0 | 4.9 | 5.4 | **5.9** | 4.6 | 5.6 | 6.0 | 9.6 | 6.7 | 3.7 | 5.2 | 5.0 | **5.8** | 5.0 | 5.7 | 6.3 | 10.0 | 6.5 | 4.1 | 5.4 | 5.3 | **6.0** |

## C.4 SENSITIVITY ANALYSIS ON TASK LOSS COEFFICIENTS

We explore the sensitivity of task loss coefficients for the corrupted prediction and masked prediction tasks, where both tasks employ the self-distillation MSE loss. The loss coefficients for these tasks are initially set to 1.0, matching the scale with that of the MLM-style cross-entropy loss, which is assigned a coefficient of 2.0. Table 11 summarizes the result of our experiments varying these loss coefficients within the CAV2vec framework.

Our observation indicates that the model performs consistently well across different coefficient settings. Meanwhile, it is recommended that the masked prediction loss coefficient is not set lower than the coefficients for ACP and VCP tasks. The masked prediction task plays a crucial role in contextualized representation learning. We also observe that when $\lambda_{\text{ACP}} = 1.0$ and $\lambda_{\text{VCP}} = 0.5$, the model outperforms particularly in conditions with low audio noise level. This result may be attributed to the informative nature of the audio target, as similarly reported in Haliassos et al. (2024), where stronger audio guidance than visual guidance is utilized for training the ASR model. However, this configuration has resulted in suboptimal performance under more challenging conditions, such as high noise levels or unseen DEMAND noise types.

Table 11: Sensitivity analysis on task loss coefficients, exploring different values for the ACP, VCP, and masked prediction tasks under various noise conditions. We present N-WER for each noise type as well as clean WER results. Visual corruption type is used as object occlusion + noise.

| $\lambda_{\text{ACP}}$ | $\lambda_{\text{VCP}}$ | $\lambda_{\text{Mask}}$ | Babble | Speech | Music | Natural | Clean |
|---|---|---|---|---|---|---|---|
| 0.5 | 0.5 | 1.0 | 9.2 | 3.2 | 4.0 | 4.0 | 1.5 |
| 0.5 | 1.0 | 1.0 | 9.0 | 3.2 | 4.0 | 3.7 | 1.5 |
| 1.0 | 0.5 | 1.0 | 9.0 | 3.1 | 3.9 | 3.9 | 1.4 |
| 1.0 | 1.0 | 1.0 | 9.2 | 3.2 | 4.1 | 3.9 | 1.5 |
| 1.0 | 2.0 | 1.0 | 9.3 | 3.2 | 4.2 | 4.1 | 1.6 |
| 2.0 | 1.0 | 1.0 | 9.1 | 3.2 | 3.9 | 3.9 | 1.6 |
| 2.0 | 2.0 | 1.0 | 9.2 | 3.3 | 4.0 | 3.9 | 1.6 |
| 1.0 | 1.0 | 0.5 | 9.1 | 3.1 | 4.1 | 4.0 | 1.6 |
| 1.0 | 1.0 | 1.0 | 9.2 | 3.2 | 4.1 | 3.9 | 1.5 |
| 1.0 | 1.0 | 2.0 | 9.4 | 3.2 | 4.1 | 3.9 | 1.6 |

## C.5 ASR AND VSR RESULTS

Completely missing modalities are common in real-world scenarios, and evaluating the single-modality performance, *i.e.,* ASR and VSR, under corrupted conditions is also essential. In Table 12, we evaluated ASR and VSR tasks, comparing three models: AV-HuBERT, AV-RelScore, and CAV2vec. For ASR, we used audio corruptions such as babble, speech, music, and natural noises (N-WER), and for VSR, we used object occlusion with noise, hands occlusion, and pixelated face. We also report results in clean conditions. The results confirm that our CAV2vec framework robustly works with both unimodal and multimodal data across all conditions, demonstrating its effectiveness in producing robust representations even when modalities are partially or completely corrupted.

Table 12: ASR and VSR task results on the LRS3 benchmark.

| Method | ASR | | | | | VSR | | | |
|---|---|---|---|---|---|---|---|---|---|
| | Babble | Speech | Music | Natural | Clean | Object | Hands | Pixelate | Clean |
| AV-HuBERT | 35.8 | 22.6 | 13.9 | 12.8 | 1.6 | 34.9 | 37.2 | 35.6 | 28.7 |
| AV-RelScore | 36.2 | 22.5 | 15.0 | 13.5 | 1.7 | 34.2 | 37.7 | 36.0 | 28.6 |
| **CAV2vec** | 35.2 | 20.3 | 13.6 | 12.3 | 1.5 | 33.9 | 36.6 | 35.6 | 27.9 |

## C.6    PRETRAINING FROM DIFFERENT INITIALIZATIONS

In our experiments, we have utilized pretrained AV-HuBERT model to train CAV2vec, since fully training an audio-visual representation learning model from scratch requires substantial resources. For instance, training AV-HuBERT using LRS3 (433h) + VoxCeleb2 (1326h) requires 600K training steps on 64 GPUs, which spans ~4 days to complete (Shi et al., 2022a). Given our limited resources, conducting full pretraining from scratch has not been feasible. Thus, the use of pre-existing weights allowed us to achieve robust performance with minimal additional training cost, demonstrating the efficiency of our method in achieving robustness with fewer resources.

Nevertheless, to investigate the effect of initialization, we compared different pretraining strategies, AV-data2vec and CAV2vec, within our resource constraints. This includes the performance of models with random initialization, pretraining on LRS3 (433h) for 120K steps with 4 GPUs, using 4 forward passes per update step to compensate for the small batch size. Table 13 shows that CAV2vec pretraining with corrupted prediction significantly outperforms AV-data2vec. Furthermore, the performance enhancement gap becomes more pronounced when the models are trained from scratch, implying the potential benefits of our method if more extensive resources were available.

Table 13: Comparison between the models pretrained from different initializations.

| Method | Unlabeled hrs | Init. | O/MS | O/DM | H/MS | H/DM | P/MS | P/DM |
|---|---|---|---|---|---|---|---|---|
| AV-data2vec | 433h | random | 10.5 | 8.5 | 11.1 | 9.0 | 10.0 | 8.2 |
| **CAV2vec** | 433h | random | 8.8 | 7.7 | 9.0 | 7.8 | 8.9 | 7.4 |
| AV-data2vec | 433h + 1326h | pretrained | 6.2 | 5.1 | 6.5 | 5.5 | 6.0 | 4.9 |
| **CAV2vec** | 433h + 1326h | pretrained | 5.1 | 4.3 | 5.2 | 4.3 | 5.1 | 4.2 |

## C.7    COMPARISON BETWEEN SELF-SUPERVISED PRETRAINING WITH CORRUPTED DATA

In the framework of CAV2vec, we designed the corrupted prediction tasks to highlight the importance of robust representation learning, and demonstrated the advantages of a unimodal strategy in corrupted representation learning. To dissect the impact of data corruption and CAV2vec's training by corrupted prediction tasks, other SSL methods could also be trained on corrupted data during the representation learning. We conducted uptraining within the AV-HuBERT and AV-data2vec pretraining frameworks, using the same data corruptions and same number of training steps as CAV2vec. As demonstrated in Table 14, the incorporation of corrupted prediction tasks in CAV2vec is critical in robust representation learning with corrupted data, highlighting its effectiveness compared to other methods which do not gain large robustness through corrupted data augmentation.

Table 14: Comparison between different self-supervised pretraining frameworks under corrupted environments. When pretraining (PT) with corrupted data, we use the same training and data configurations across all models.

| Method | PT w/ corrupted data | O/MS | O/DM | H/MS | H/DM | P/MS | P/DM |
|---|---|---|---|---|---|---|---|
| AV-HuBERT | ✗ | 6.0 | 5.0 | 6.2 | 5.4 | 5.8 | 4.9 |
| AV-data2vec | ✗ | 6.2 | 5.1 | 6.5 | 5.5 | 6.0 | 4.9 |
| AV-HuBERT | ✓ | 5.8 | 4.9 | 5.9 | 5.1 | 5.8 | 4.7 |
| AV-data2vec | ✓ | 5.8 | 4.9 | 6.0 | 5.0 | 5.9 | 4.8 |
| **CAV2vec** | ✓ | 5.1 | 4.3 | 5.2 | 4.3 | 5.1 | 4.2 |

