# OpenReview forum: "Multi-Task Corrupted Prediction for Learning Robust Audio-Visual Speech Representation"
_ICLR.cc/2025/Conference — ICLR 2025 Poster_

### Official Review · Reviewer_Vqmw · 2024-11-03

**Soundness:** 2
**Presentation:** 3
**Contribution:** 2
**Rating:** 5
**Confidence:** 4

**Summary:**

This paper proposes a self-distillation framework for audio-visual speech representation learning. It enhances representation robustness through multimodal and unimodal multi-task learning. Experimental results demonstrate performance improvements over existing methods.

**Strengths:**

+ The target is clear. It aims to handle audio-visual joint corruption in AVSR, which is quite common in real-world.

+ The method is simple and efficient, requiring no additional modules to handle audio-visual corruptions, which makes it practical for implementation.

**Weaknesses:**

- The work is similar to several existing works in the view of methodology, like AV-Hubert, raven, dinoSR, ES^3. One difference is the cross-modal prediction of both A→V and V→A, instead of just V→A, which is mainly due to the introduction of corruptions in both A and V modalities in this work. Although the starting point of tackling corruptions in both modalities is meaningful, this specific learning manner is not so insightful or so original. Compared with those existing works, there are little new things in this work. There are already many works learning audio-visual speech representations with EMA framework at present. I am not clear about the novelty of the work in the view of methodology compared with existing works.

- The validation in this paper is not thorough enough, as the authors only demonstrated improvements on the AVSR task, although they explicitly stated their focus on AVSR. In real-world scenarios, complete corruption of one modality is possible, and missing modalities are common. If the audio-visual representations learned by this method are truly robust, then performance on single-modality tasks (ASR and VSR) would help substantiate this claim.

- The reliance on AV-HuBERT pre-trained weights raises questions about the framework's standalone capabilities. It remains unclear whether comparable performance could be achieved training from scratch, or if robust audio-visual representations are inherently dependent on AV-HuBERT.

- The manuscript suffers from clarity issues and leaves several questions (detailed below).

**Questions:**

1. Figure 3's notation lacks precision. While the bidirectional arrows suggest symmetric interactions, the prediction task implies directed objectives.

2. What is the definition of "visual corruption events"? Is it having a single corrupted frame in the whole visual sequence or something else? In Appendix A.2, the description "Object occlusion applied to the speaker's lips occurs once, followed by Gaussian noise or blurring, each with a probability of 0.3", does this mean that occlusion only appears in a single frame within the sequence?

3. The rationale for applying MP (Masked Prediction) exclusively in multimodal settings, while CP (Corruption Prediction) spans both multimodal and unimodal scenarios. It would be better to clarify the reasons.

4. The parameter k in MP's objective (targeting the average of teacher model's top-k blocks output) is not specified, and the motivation behind this design is not explained.

5. The absence of L_AVCP in the overall multi-task loss function requires explanation.

6. What is the motivation for MLM and what is the setting of it? How to acquire the cluster index? Need explanation.

---

> ### Author Response · Authors · 2024-11-19
> **Author Response to Reviewer Vqmw (Part 1)**
>
> > Weakness 1: The work is similar to several existing works in the view of methodology, like AV-Hubert, raven, dinoSR, ES^3. One difference is the cross-modal prediction of both A→V and V→A, instead of just V→A, which is mainly due to the introduction of corruptions in both A and V modalities in this work. Although the starting point of tackling corruptions in both modalities is meaningful, this specific learning manner is not so insightful or so original. Compared with those existing works, there are little new things in this work. There are already many works learning audio-visual speech representations with EMA framework at present. I am not clear about the novelty of the work in the view of methodology compared with existing works.
>
> A1: Thank you for your insights and the opportunity to further clarify the unique aspects and contributions of our approach.
> The key distinction of CAV2vec lies in its incorporation of corrupted prediction tasks into speech representation learning, particularly emphasizing the effectiveness of unimodal multi-task strategies for achieving robustness. Our work is the first to explicitly leverage and explore robust representation learning strategies to address audio-visual joint corruption.
>
> While one might consider simply extending models like AV-data2vec or DinoSR with corrupted data, such a straightforward extension fails to address critical challenges. For instance, in AVSR environments, corruption often disrupts the synchronization between audio and video modalities. Using mixed audio-visual information for both inputs and targets can obscure which modality the corruption affects, making it difficult to isolate and address corruption effectively.
>
> Instead, CAV2vec introduces a novel strategy by employing unimodal corrupted prediction tasks alongside conventional multimodal masked predictions. This approach is distinct from other audio-visual models like AV-HuBERT, AV-data2vec, or ES$^3$, which do not differentiate roles between multimodal and unimodal data during training. Furthermore, models like RAVEn, BRAVEn, and DinoSR, while effective in clean ASR, do not employ an audio-video fusion to form robust multimodal features, which is essential in scenarios where both modalities may be corrupted.
>
> In Table 4, we have found that unimodal and cross-modal corrupted prediction (ACP, VCP) is significantly more effective than using multimodal data as corrupted input features or targets (AVCP, mACP, mVCP). The reliability of multimodal fusion has also been enhanced, substantiated by the similarity measure between sample sequence features in Figure 4.
>
> We will further elaborate the detailed distinction of CAV2vec in the related work section. Also, we recognize the importance of existing models such as DinoSR and ES$^3$, and we will include them in our analysis to offer a thorough comparison and review.
>
> > Weakness 2: The validation in this paper is not thorough enough, as the authors only demonstrated improvements on the AVSR task, although they explicitly stated their focus on AVSR. In real-world scenarios, complete corruption of one modality is possible, and missing modalities are common. If the audio-visual representations learned by this method are truly robust, then performance on single-modality tasks (ASR and VSR) would help substantiate this claim.
>
> A2: Thank you for your valuable comments on single-modality tasks evaluation. Recognizing that missing modalities are common in real-world scenarios, we agree that evaluating ASR and VSR performance under corrupted conditions are also essential. Consequently, we evaluated ASR and VSR tasks, comparing three models: AV-HuBERT, AV-RelScore, and CAV2vec. For ASR, we used audio corruptions such as babble, speech, music, and natural noises (N-WER); for VSR, we used object occlusion with noise, hands occlusion, and pixelate.
>
> | Method      | ASR (Babble / Speech / Music / Natural / Clean) | VSR (Object / Hands / Pixelate / Clean) |
> |-------------|-------------------------------------------------|----------------------------------------|
> | AV-HuBERT   | 35.8 / 22.6 / 13.9 / 12.8 / 1.6                | 34.9 / 37.2 / 35.6 / 28.7              |
> | AV-RelScore | 36.2 / 22.5 / 15.0 / 13.5 / 1.7                | 34.2 / 37.7 / 36.0 / 28.6              |
> | **CAV2vec**     | **35.2 / 20.3 / 13.6 / 12.3 / 1.5**                | **33.9 / 36.6 / 35.6 / 27.9**              |
>
> The results, as shown in the table, confirm that our CAV2vec framework robustly works with both unimodal and multimodal data across all conditions, demonstrating its effectiveness in producing robust representations even when modalities are partially or completely corrupted. We added this result in Appendix C.5.

---

> ### Author Response · Authors · 2024-11-19
> **Author Response to Reviewer Vqmw (Part 2)**
>
> > Weakness 3: The reliance on AV-HuBERT pre-trained weights raises questions about the framework's standalone capabilities. It remains unclear whether comparable performance could be achieved training from scratch, or if robust audio-visual representations are inherently dependent on AV-HuBERT.
>
> A3: In the table below, we compared different pretraining strategies, AV-data2vec and CAV2vec, within our resource limits. This includes the performance of models with random initialization, pretraining on LRS3 (433h) for 120K steps with 4 GPUs (using 4 forward passes per update step).
>
> | Method         | O/MS | O/DM | H/MS | H/DM | P/MS | P/DM |
> |----------------|------|------|------|------|------|------|
> | AV-data2vec (random init.) | 10.5 |  8.5 | 11.1 |  9.0 | 10.0 |  8.2 |
> | CAV2vec (random init.)     |  8.8 |  7.7 |  9.0 |  7.8 |  8.9 |  7.4 |
>
> It is evident that CAV2vec pretraining with corrupted prediction significantly outperforms AV-data2vec. Furthermore, the performance enhancement gap becomes more pronounced when the models are trained from scratch, implying the potential benefits of our method if more extensive resources were available.
>
> However, fully training an audio-visual representation learning model from scratch requires substantial resources. For instance, training AV-HuBERT using LRS3 + VoxCeleb2 (1759h) requires 600K training steps on 64 GPUs which spans ~four days to complete. Given our limited resources, conducting full pretraining from scratch was not feasible. Thus, the use of pre-existing weights allowed us to achieve robust performance with minimal additional training cost, demonstrating the efficiency of our method in achieving robustness with fewer resources.
>
> > Q1: Figure 3's notation lacks precision. While the bidirectional arrows suggest symmetric interactions, the prediction task implies directed objectives.
>
> A4: Thank you for pointing out the potential confusion in Figure 3. We revised the arrows to unidirectional ones.
>
> > Q2: What is the definition of "visual corruption events"? Is it having a single corrupted frame in the whole visual sequence or something else? In Appendix A.2, the description "Object occlusion applied to the speaker's lips occurs once, followed by Gaussian noise or blurring, each with a probability of 0.3", does this mean that occlusion only appears in a single frame within the sequence?
>
> A5: Visual corruption event is not defined based on a single-frame corruption; instead, it refers to the number of times contiguous frames are corrupted as a chunk within a sequence. The number of contiguous frames corrupted is defined as "corruption length", which for visual corruptions ranges 10–50% of the sequence and for audio corruptions ranges 30–50% of the sequence.
>
> For example, if there are two visual corruption events within a sequence, one could be an object occlusion affecting 30% (ratio may randomly change) of the sequence, and the other could be Gaussian noise affecting another 30% of the sequence. Similarly, if three hand occlusion events occur, it means that three separate chunks of the sequence are corrupted using three different hand images.
>
> > Q3: The rationale for applying MP (Masked Prediction) exclusively in multimodal settings, while CP (Corruption Prediction) spans both multimodal and unimodal scenarios. It would be better to clarify the reasons.
>
> A6: In this study, our primary contribution is not to develop novel contextualized representation learning but to explore robust representation learning methods that effectively handle corrupted audio-visual data. Therefore, we focused on various strategies for corrupted prediction tasks, while following the standard AV-data2vec approach for the masked prediction task. Additionally, we experimented with incorporating unimodal masked prediction loss into the CAV2vec framework; however, this modification did not significantly impact performances (9.2 / 3.3 / 4.1 / 3.9 / 1.5 % under babble / speech / music / natural / clean with visual object occlusion).
>
> > Q4: The parameter k in MP's objective (targeting the average of teacher model's top-k blocks output) is not specified, and the motivation behind this design is not explained.
>
> A7: The use of top-$k$ blocks, where we normalize and average the outputs of the top-$k$ blocks to use as the target, follows the implementation from data2vec and AV-data2vec. This approach has been shown to yield effective results by integrating diverse outputs from multiple layers, resulting in smoother and higher-quality targets, and preventing representation collapse. In our internal experiments, we observed that $k \ge 8$ consistently yielded similarly good results.

---

> ### Author Response · Authors · 2024-11-19
> **Author Response to Reviewer Vqmw (Part 3)**
>
> > Q5: The absence of L_AVCP in the overall multi-task loss function requires explanation.
>
> A8: $\mathcal{L}_\text{AVCP}$ initially was suggested as an extension of masked prediction loss of AV-data2vec to a corrupted prediction. Although this approach proved effective, there is a limitation with multimodal predictions in addressing the corruptions on individual modalities, which makes us employ only unimodal corrupted prediction tasks (ACP + VCP) as outlined in Eq.(4).
>
> Our experiments have demonstrated that using unimodal targets for corrupted prediction tasks yields better results than using multimodal targets (refer to Table 4: mACP + mVCP + AVCP). Consequently, the final objective function of CAV2vec exclusively employs unimodal corrupted prediction tasks, while multimodal targets are reserved for the original masked prediction tasks. For additional information, we included the result of using ACP + VCP + AVCP tasks in Table 4, which also presents subpar performance.
>
>
> > Q6: What is the motivation for MLM and what is the setting of it? How to acquire the cluster index? Need explanation.
>
> A9: As discussed in L.308-310, several approaches for speech representation, including AV-HuBERT and AV-data2vec, employ MLM loss to promote faster convergence. We have adopted a similar approach, using cross-entropy loss to predict the cluster index of each frame’s features. In addition, we adjust the weight $\lambda_\text{MLM}$ to integrate MLM with self-distillation loss. To acquire cluster indices, we pass all training data through a pretrained AV-HuBERT model and then apply k-means clustering to the features. This clustering approach can be implemented with any publicly available pretrained model.

---

> ### Comment · Area_Chair_gd3h · 2024-11-24
>
> Dear Reviewer Vqmw,
>
> Thank you for your efforts and contributions as a reviewer. As we approach the final two days of the discussion phase, I noticed that your response to the authors' rebuttal is still pending.
>
> Your feedback is crucial to facilitate consensus among reviewers, especially since this paper has received a set of divergent reviews. Could you kindly provide your thoughts based on the authors' rebuttal? Additionally, if there are any potential changes to your final rating, it would be helpful to explicitly mention them in your comment.
>
> Best regards,
> AC

---

> ### Author Response · Authors · 2024-12-02
> **Gentle Reminder to Reviewer Vqmw**
>
> Dear Reviewer Vqmw,
>
> Thank you again for your time and effort in reviewing our paper. We greatly appreciate your valuable feedback and suggestions.
>
> We’d like to gently remind you that the discussion period is coming to an end.
>
> In our rebuttal, we addressed your concerns by:
>
> - **Clarifying the contributions of our study,**
> - **Supplementing our results with additional ASR and VSR tasks,**
> - **Detailing the findings from experiments with random-initialized models,**
> - **Addressing several related questions and clarifying the details.**
>
> If you have any remaining concerns, please do not hesitate to share them with us. We are more than willing to address them promptly. Thank you very much for your consideration.
>
> Best regards,
> Authors

---

### Official Review · Reviewer_ETWb · 2024-11-04

**Soundness:** 3
**Presentation:** 3
**Contribution:** 3
**Rating:** 8
**Confidence:** 5

**Summary:**

This paper presents a self-supervised framework for learning audio-visual speech representations which are robust to noise. This is achieved by including audio and visual corruptions during the self-supervised learning (SSL) phase. As a consequence, the model is more robust to audio and visual noise during evaluation. Results are shown on the LRS2 and LRS3 datasets and the proposed approach outperforms prior SSL approaches on noisy conditions.

**Strengths:**

The paper is well written.

The paper investigates the use of audio and visual corruptions during the SSL phase which has not been thoroughly investigated in previous works.

**Weaknesses:**

The paper has a lot of similarities with AV-data2vec, AV-Hubert, Raven and Braven, especially regarding the masked prediction and cross-modal prediction tasks. It would be very useful if some discussion is added in the related work section explaining the differences with each of these approaches.

Since the proposed model is pre-trained with corrupted data it's reasonable that it performs better than the other SSL methods which have not been trained with such data. The authors should make clear what the contribution is. If it's just the addition of corruptions during pre-training and then it should be clearly stated that the other approaches might also benefit if they are pre-trained with such corruptions.

The proposed model is initialised with AV-HuBERT weights. This is done for efficiency but it’s still different from other methods which are randomly initialised. Can the authors comment on the impact of this initialisation? Are there any results where the model is trained from scratch?

The number of parameters in Table 1 is not 100% accurate. Raven and Braven have 673M during pre-training and during inference, however the other approach have 325M only during the pre-training phase. During inference the number of parameters is double, since models are separately fine-tuned for each modality.

L. 351-353 state that publicly available pre-trained models  are used and then fine-tune them based on their hyperparameter settings. This leads to significantly worse results for RAVEn / BRAVEn than the ones published. For example, on clean LRS3 audio, results of 2.3 / 1.8 % WER for RAVEn / BRAVEn are presented whereas the published results are 1.4 / 1.1 % WER. Can the authors comment on this discrepancy? And maybe explain why this happens?

L.18: Does the model predict “corrupted frames” as stated? I think it predicts clean frames.

**Questions:**

Please see above.

---

> ### Author Response · Authors · 2024-11-19
> **Author Response to Reviewer ETWb (Part 1)**
>
> > Weakness 1: The paper has a lot of similarities with AV-data2vec, AV-Hubert, Raven and Braven, especially regarding the masked prediction and cross-modal prediction tasks. It would be very useful if some discussion is added in the related work section explaining the differences with each of these approaches.
>
> A1: Thank you for your insights and comparisons to existing works in the field of audio-visual speech representation learning. We appreciate the opportunity to clarify and highlight the unique aspects and contributions of our approach.
>
> The key distinction of CAV2vec lies in its incorporation of corrupted prediction tasks into speech representation learning, particularly emphasizing the effectiveness of unimodal multi-task strategies for achieving robustness. Our work is the first to explicitly leverage and explore robust representation learning strategies to address audio-visual joint corruption.
>
> To achieve this, CAV2vec builds upon a masked prediction/modeling framework for learning contextualized speech representations, similar to AV-data2vec (distillation) and AV-HuBERT (cluster prediction), enhancing it with a corrupted prediction framework for robustness. Besides, while (B)RAVEn employs cross-modal prediction to jointly train two unimodal models and primarily harnesses informative audio targets, CAV2vec trains to handle corruption in both audio and video modalities simultaneously, ensuring robust and reliable multimodal fusion.
>
> We will further elaborate the detailed distinction and the specific contributions of CAV2vec in the related work section.
>
>
> > Weakness 2: Since the proposed model is pre-trained with corrupted data it's reasonable that it performs better than the other SSL methods which have not been trained with such data. The authors should make clear what the contribution is. If it's just the addition of corruptions during pre-training and then it should be clearly stated that the other approaches might also benefit if they are pre-trained with such corruptions.
>
> A2: In this study, the main contributions are: (1) designing the corrupted prediction tasks to highlight the importance of robust representation learning, and (2) demonstrating the advantages of a unimodal strategy in corrupted representation learning. To dissect the impact of data corruption and CAV2vec’s training by corrupted prediction tasks, other SSL methods could also be trained on corrupted data during the representation learning. We conducted uptraining within the AV-HuBERT and AV-data2vec pretraining frameworks, using the same data corruptions and same number of steps as CAV2vec.
>
> As demonstrated in the table below, the incorporation of corrupted prediction tasks in CAV2vec is critical in robust representation learning with corrupted data, highlighting its effectiveness compared to other methods. Please note that every model in the table has been fine-tuned under identical corruption configurations (L.366-367).
>
> | Method       | Pretrain with corrupted data | O/MS | O/DM | H/MS | H/DM | P/MS | P/DM |
> |--------------|------------------------------|------|------|------|------|------|------|
> | AV-HuBERT    | x                            | 6.0  | 5.0  | 6.2  | 5.4  | 5.8  | 4.9  |
> | AV-data2vec  | x                            | 6.2  | 5.1  | 6.5  | 5.5  | 6.0  | 4.9  |
> |  =========== | ========================== |  ===  | ===  |  ===  | === |  === |  ===  |
> | AV-HuBERT    | o                            | 5.8  | 4.9  | 5.9  | 5.1  | 5.8  | 4.7  |
> | AV-data2vec  | o                            | 5.8  | 4.9  | 6.0  | 5.0  | 5.9  | 4.8  |
> | **CAV2vec**      | o                            | **5.1**  | **4.3**  | **5.2**  | **4.3**  | **5.1**  | **4.2**  |

---

> ### Author Response · Authors · 2024-11-19
> **Author Response to Reviewer ETWb (Part 2)**
>
> > Weakness 3: The proposed model is initialized with AV-HuBERT weights. This is done for efficiency but it’s still different from other methods which are randomly initialized. Can the authors comment on the impact of this initialization? Are there any results where the model is trained from scratch?
>
> A3: In the table below, we compared different pretraining strategies, AV-data2vec and CAV2vec, within our resource limits. This includes the performance of models with random initialization, pretraining on LRS3 (433h) for 120K steps with 4 GPUs (using 4 forward passes per update step).
>
> | Method         | O/MS | O/DM | H/MS | H/DM | P/MS | P/DM |
> |----------------|------|------|------|------|------|------|
> | AV-data2vec (random init.) | 10.5 |  8.5 | 11.1 |  9.0 | 10.0 |  8.2 |
> | CAV2vec (random init.)     |  8.8 |  7.7 |  9.0 |  7.8 |  8.9 |  7.4 |
>
> It is evident that CAV2vec pretraining with corrupted prediction significantly outperforms AV-data2vec. Furthermore, the performance enhancement gap becomes more pronounced when the models are trained from scratch, implying the potential benefits of our method if more extensive resources were available.
>
> However, fully training an audio-visual representation learning model from scratch requires substantial resources. For instance, training AV-HuBERT using LRS3 + VoxCeleb2 (1759h) requires 600K training steps on 64 GPUs which spans ~four days to complete. Given our limited resources, conducting full pretraining from scratch was not feasible. Thus, the use of pre-existing weights allowed us to achieve robust performance with minimal additional training cost, demonstrating the efficiency of our method in achieving robustness with fewer resources.
>
> > Weakness 4: The number of parameters in Table 1 is not 100% accurate. Raven and Braven have 673M during pre-training and during inference, however the other approach have 325M only during the pre-training phase. During inference the number of parameters is double, since models are separately fine-tuned for each modality.
>
> A4: In Table 1, we provide the number of parameters of the encoders alone. In our experiments, all models utilize the same 9-layer Transformer decoder for inference, and thus the comparison focuses on the encoder parameters. RAVEn, BRAVEn, AV-HuBERT, AV-data2vec, and CAV2vec all use encoders based on a similar 24-layer Transformer architecture. However, RAVEn and BRAVEn have ~2x parameters than the others because they are basically unimodal models for distinct ASR and VSR tasks. To perform AVSR, both models require two separate 24-layer encoders—the approach of combining features from two pretrained encoders to feed into the decoder is elaborated in the BRAVEn paper.
>
> We have rechecked the number of parameters for each encoder and confirmed the numbers in Table 1 were correct.
>
> > Weakness 5: L.351-353 state that publicly available pre-trained models are used and then fine-tune them based on their hyperparameter settings. This leads to significantly worse results for RAVEn / BRAVEn than the ones published. For example, on clean LRS3 audio, results of 2.3 / 1.8 % WER for RAVEn / BRAVEn are presented whereas the published results are 1.4 / 1.1 % WER. Can the authors comment on this discrepancy? And maybe explain why this happens?
>
> A5: The discrepancies in WER results can be attributed to different evaluation settings. Our settings are designed to assess noise-robust audio-visual models under real-world conditions, where the type and extent of modality corruption are unpredictable. Our WER 2.3 / 1.8% report performances under visual corruption, whereas the published results pertain to clean audio conditions using a standalone ASR model. Although standalone ASR models excel in clean audio environments, their performance significantly degrades under noisy conditions.
>
> BRAVEn has reported low-resource AVSR performance (WER 4.7 / 4.0%), but it does not provide results in high-resource settings or under audio-visual corruption. Moreover, the specific hyperparameters used for fine-tuning in conjunction with pretrained ASR and VSR encoders and a decoder are not detailed in their publication.
>
> The performance gap might also stem from the larger unlabeled dataset and self-training technique during pretraining and the use of a language model during inference, which were not employed in our experimental setup. Also, while (B)RAVEn used CTC/attention loss for fine-tuning, we only used attention loss to ensure a fair comparison across all models. These variations in decoding approaches can influence outcomes, although they could be orthogonally applied to any models.
>
> > Weakness 6: L.18: Does the model predict “corrupted frames” as stated? I think it predicts clean frames.
>
> A6: Thank you for your clarification. You are correct; the model predicts clean targets for corrupted input frames. We revised the manuscript accordingly.

---

> > ### Comment · Reviewer_ETWb · 2024-11-26
> >
> > I would like to thank the authors for their detailed reply.
> >
> > Please add the relevant discussion for A1 to the paper.
> >
> > Please add the results and related discussion for A2 and A3 to the supplementary material.
> >
> > Please clearly explain in the paper why there are discrepancies in the numbers (A5).

---

> > > ### Author Response · Authors · 2024-12-02
> > > **Official Comment by Authors**
> > >
> > > Thank you for updating the score. We are glad that we have dismissed all your concerns via the rebuttal.
> > >
> > > And yes, we will make sure to add/clarify every detail in the following revision. Thank you again for your valuable comments.

---

### Official Review · Reviewer_ioDE · 2024-11-05

**Soundness:** 3
**Presentation:** 2
**Contribution:** 3
**Rating:** 8
**Confidence:** 2

**Summary:**

The paper introduces a technique called CAV2vec which is a model trained with a so-called corrupted prediction task: the student's encoded output on corrupted input is trained to be similar to the teacher's output on clean (uncorrupted) input in an audio-visual speech recognition task consisting of audio (speech) and video (lip tracks) inputs. They specifically using cross-modal targeting gives additional benfit. The demonstrate success against a variety of strong models, yielding equivalent state-of-the-art for AVSR on uncorrupted input (vs AV data2vec) and better results than state-of-the-art on artificially corrupted data.

**Strengths:**

Paper shows good results against a variety of relevant, strong baselines. In table 1 they show that on both visual and audio corruption CAV2vec shows the best robustness. In some ways, this could be considered an extension of data2vec with the corrupted prediction task yielding better results. In this light, it is notable that the task does not degrade on the Clean condition (e.g. 1.5% is still the best number in Table 1). However on the artificially corrupted noisy conditions, Cav2vec is peforming more than 10% relative on average than the next best systems. The paper also provides ablations to tease out what part of the corruption is helping most, e.g. ACP + VCP.

**Weaknesses:**

The clearest wins are on the artificially created corrupted noise sets---this is often the case when the experiments simulate harder test conditions and corresponding training conditions to address the harder test. Other systems, do not have the benefit of training to match the altered test conditions.

The various acronyms AVCP, mACP, mCP, ACP, VCP are somewhat hard to follow. While Figure 5 helps, perhaps a table indicating exactly what they mean with columns indicating differing dimensions could help (it may not...).

**Questions:**

Are there harder, non-simualted noisier datasets that could be used to evaluated the benefit of CAV2vec over say data2vec?

Is the data2vec system trained on, is it possible to train on, the same distributions and corruptions in training as the CAV2vec system?

---

> ### Author Response · Authors · 2024-11-19
> **Author Response to Reviewer ioDE**
>
> > Weakness 1: The clearest wins are on the artificially created corrupted noise sets---this is often the case when the experiments simulate harder test conditions and corresponding training conditions to address the harder test. Other systems, do not have the benefit of training to match the altered test conditions.
>
> A1: We note that CAV2vec and all other baseline models have been fine-tuned under identical corruption configurations (L.366-367). Thus, CAV2vec's enhanced performance is not due to specific training to match test conditions but rather its advanced capacity for producing robust representation, which is a key contribution of our work. Other baselines, even when fine-tuned on corrupted data, struggle to produce such robust representations.
>
> Additionally, to simulate and assess real-world scenarios where training and test conditions differ, our study has introduced unseen corruption types during evaluation, which are not presented during training. These unseen types (Figure 2), e.g., hands occlusion or pixelated face, and DEMAND audio noise, simulate unexpected corruptions that models may encounter in practice. Please also refer to the results of our ASR and VSR task detailed in Appendix C.5, where we assume that completely corrupted or missing modality is common in the real-world.
>
> > Weakness 2: The various acronyms AVCP, mACP, mCP, ACP, VCP are somewhat hard to follow. While Figure 5 helps, perhaps a table indicating exactly what they mean with columns indicating differing dimensions could help (it may not...).
>
> A2: Thank you for your feedback on the use of acronyms in our paper. To alleviate the confusion, we summarized the specific notations in Table 6 in Appendix C.1, and will also enhance Figure 5 to clarify the meanings of each acronym. Specifically, we use the prefix 'm-' to denote multimodal input features. For the corrupted prediction tasks, 'A' in ACP represents the use of an audio-only target, 'V' represents a video-only target, and 'AV' represents an audio-visual target.
>
> > Q1: Are there harder, non-simualted noisier datasets that could be used to evaluated the benefit of CAV2vec over say data2vec?
>
> A3: Common benchmarks in AVSR, such as LRS2 and LRS3, typically provide curated datasets that have refined or removed low-quality data, featuring clear audio and visible speakers. Consequently, to simulate real-world challenges, we needed to synthetically introduce corruptions into our dataset. Similar setup has been employed in other studies on audio-visual corruption [1, 2]. For this reason, as previously mentioned, we also designed our experimental setup with seen and unseen corruption types to assess unexpected test conditions.
>
> > Q2: Is the data2vec system trained on, is it possible to train on, the same distributions and corruptions in training as the CAV2vec system?
>
> A4: Regarding the fine-tuning phase in our experiments, both CAV2vec and other baseline models, including AV-data2vec, have been trained with the same data distribution and corruption strategies.
>
> Regarding the representation learning (uptraining phase), it is possible to train data2vec on the corrupted data as well. To dissect the impact of data corruption and learning by corrupted prediction tasks, we can augment audio-visual data with corruption during the pretraining of baseline models. Thus, we conducted this training within the AV-HuBERT and AV-data2vec pretraining frameworks, using the same data corruptions and same number of steps as CAV2vec.
>
> As demonstrated in the table below, the incorporation of corrupted prediction tasks in CAV2vec is critical in robust representation learning with corrupted data, highlighting its effectiveness compared to other methods.
>
> | Method       | Pretrain with corrupted data | O/MS | O/DM | H/MS | H/DM | P/MS | P/DM |
> |--------------|------------------------------|------|------|------|------|------|------|
> | AV-HuBERT    | x                            | 6.0  | 5.0  | 6.2  | 5.4  | 5.8  | 4.9  |
> | AV-data2vec  | x                            | 6.2  | 5.1  | 6.5  | 5.5  | 6.0  | 4.9  |
> |  =========== | ========================== |  ===  | ===  |  ===  | === |  === |  ===  |
> | AV-HuBERT    | o                            | 5.8  | 4.9  | 5.9  | 5.1  | 5.8  | 4.7  |
> | AV-data2vec  | o                            | 5.8  | 4.9  | 6.0  | 5.0  | 5.9  | 4.8  |
> | **CAV2vec**      | o                            | **5.1**  | **4.3**  | **5.2**  | **4.3**  | **5.1**  | **4.2**  |
>
> ---
> References:
>
> [1] Hong, Joanna, et al. "Watch or listen: Robust audio-visual speech recognition with visual corruption modeling and reliability scoring." Proceedings of the IEEE/CVF Conference on Computer Vision and Pattern Recognition. 2023.
>
> [2] Wang, Jiadong, et al. "Restoring Speaking Lips from Occlusion for Audio-Visual Speech Recognition." Proceedings of the AAAI Conference on Artificial Intelligence. 2024.

---

> > ### Comment · Reviewer_ioDE · 2024-11-25
> >
> > Thanks for your reply and clarifications.
> >
> > > A3: Common benchmarks in AVSR, such as LRS2 and LRS3, typically provide curated datasets that have refined or removed low-quality data, featuring clear audio and visible speakers. Consequently, to simulate real-world challenges, we needed to synthetically introduce corruptions into our dataset. Similar setup has been employed in other studies on audio-visual corruption [1, 2]. For this reason, as previously mentioned, we also designed our experimental setup with seen and unseen corruption types to assess unexpected test conditions.
> >
> > Yes, one approach to simulate real-word challenges is to introduce corruptions, ideally unseen ones. However, the best, albeit more costly to obtain, methodology is to use real-world data, even if it is in limited amount to avoid any defects that are inadvertently present in simulated corruptions.
> >
> > I don't see any new evidence that would cause me to change my initial score.

---

> ### Author Response · Authors · 2024-12-02
> **Additional Response by Authors**
>
> Thank you for the response and your valuable feedback for evaluating on the real-world dataset.
>
> In response to the reviewer’s remaining concern, we have additionally conducted evaluating the models on the Ego4D dataset [3] during the discussion period, which contains noisier and corrupted, real-world speech videos. The data frequently feature missing speaker face or ambiguous visions in the camera view, and the speech audio is often unclear.
>
> We evaluated the models without any synthetic audio-visual corruption introduced. The results (WER) are as follows: AV-HuBERT 76.4%, AV-data2vec 77.0%, AV-RelScore 76.7%, and CAV2vec 74.1%, which demonstrates the CAV2vec's enhancements in real-world corruption scenarios.
>
> Thank you again for your time and effort in reviewing our paper. We believe that your feedback has significantly improved the contributions of our study.
>
> ---
> [3] Grauman, Kristen, et al. "Ego4d: Around the world in 3,000 hours of egocentric video." Proceedings of the IEEE/CVF Conference on Computer Vision and Pattern Recognition. 2022.

---

### Official Review · Reviewer_adiV · 2024-11-11

**Soundness:** 3
**Presentation:** 3
**Contribution:** 3
**Rating:** 6
**Confidence:** 4

**Summary:**

The paper introduces CAV2vec, a self-supervised audio-visual speech representation learning framework focused on robustness to joint audio-visual corruption in noisy environments.
CAV2vec uses a self-distillation framework with corrupted prediction tasks. It leverages a teacher-student model where the student learns to predict corrupted frames while clean targets are generated by the teacher model. They proposed an unimodal multi-task learning strategy that mitigates the dispersion in the representation space caused by corrupted modalities, thereby enhancing reliable and robust audio-visual fusion.

**Strengths:**

Clear Problem Definition: The paper clearly identifies limitations in existing methods and proposes a well-reasoned solution.
Novelty: The approach of using unimodal multi-task learning for cross-modal corrupted prediction is innovative and addresses a real-world challenge in AVSR.
Technical Depth: The framework uses a combination of advanced techniques, including self-distillation, multimodal Transformer encoders, and self-supervised learning without architectural changes.
Empirical Validation: The experiments on multiple benchmarks (LRS2, LRS3) and various corruption scenarios demonstrate improved robustness.
Clarity: The paper is well-structured, and technical sections are detailed with visual aids, providing clarity on the approach and results.

**Weaknesses:**

The evaluation on the LRS2 benchmark could be more comprehensive. The authors should consider including additional experimental results, similar to those presented for the LRS3 benchmark. This would better illustrate the model's robustness under real-world conditions.

**Questions:**

There is an error in the sentences on lines 421 to 422. Specifically, the description of CAV2vec’s performance on unseen types of corruption (i.e., hand occlusion and pixelated face) is incorrect. According to Table 1, the correct values should be 5.2 and 5.1, respectively.

---

> ### Author Response · Authors · 2024-11-19
> **Author Response to Reviewer adiV**
>
> > Weakness 1: The evaluation on the LRS2 benchmark could be more comprehensive. The authors should consider including additional experimental results, similar to those presented for the LRS3 benchmark. This would better illustrate the model's robustness under real-world conditions.
>
> A1: Thank you for highlighting the need for comprehensive LRS2 benchmark results. We added the full LRS2 results for all visual and audio corruption types in Appendix C.4, similar to the detailed evaluations presented for the LRS3 benchmark.
> For a brief summary, CAV2vec decreases N-WER by up to 1.7% (object occlusion), 1.3% (hands occlusion), and 1.2% (pixelate) under MUSAN audio noise, and 0.8% (object occlusion), 1.1% (hands occlusion), 0.6% (pixelate) under DEMAND audio noise, compared to the baselines. Please review the revised appendix for full results, which demonstrate CAV2vec’s robustness under real-world conditions.
>
> > Q1: There is an error in the sentences on lines 421 to 422. Specifically, the description of CAV2vec’s performance on unseen types of corruption (i.e., hand occlusion and pixelated face) is incorrect. According to Table 1, the correct values should be 5.2 and 5.1, respectively.
>
> A2: Thank you for pointing out the error in the description of CAV2vec’s performance under hands occlusion and pixelated face. We have now revised these values in the paper to match the correct numbers as listed in Table 1.

---

> ### Author Response · Authors · 2024-12-02
> **Gentle Reminder to Reviewer adiV**
>
> Dear Reviewer adiV,
>
> Thank you again for your time and effort in reviewing our paper. We greatly appreciate your valuable feedback and suggestions.
>
> We’d like to gently remind you that the discussion period is coming to an end.
>
> In our rebuttal, we addressed your concerns by:
> - **Supplementing our comprehensive results with the LRS2 benchmark,**
> - **Revising the errors for results in the manuscript.**
>
> If you have any remaining concerns, please do not hesitate to share them with us. We are more than willing to address them promptly. Thank you very much for your consideration.
>
> Best regards,
> Authors

---

### Author Response · Authors · 2024-11-24
**General Response by Authors**

Dear all reviewers,

Thank you for the time and effort you have dedicated to reviewing our paper and providing such valuable feedback. Your insights have been instrumental in enhancing the quality of our paper. In response to your comments, we have uploaded a detailed response for each review and revised the manuscript accordingly, particularly by supplementing it with new experimental results during the rebuttal phase.

We would like to gently remind the reviewers that the discussion period will close in two days. We are committed to addressing any further clarifications or questions you may still have. Thank you once again for your invaluable contributions to improving our work.

Best regards,
Authors

---

### Meta-Review · Area_Chair_gd3h · 2024-12-20

**Metareview:**

The paper received three positive reviews (two 8s and a 6) and a negative review (5). The paper introduces a novel method for AVSR with joint AV corruption incorporating self-distillation. The reviews mainly requested additional clarifications, experiments and discussions that would potentially strengthen the paper and the author addressed most of them. While the negative reviewer requested clarifications on the novelty of the work and additional experiments, which the authors provided in their rebuttal, the review did not responded to the author rebuttal. The AC believes that the author response largely resolved the issues raised by the reviewer and therefore recommends the acceptance of the paper following the majority of the reviewer's ratings.

**Additional Comments On Reviewer Discussion:**

The reviews mainly requested additional experiments and discussions as well as some clarifications. The authors provided answers with additional results, which most reviewers acknowledged for. A single negative reviewer did not provide their response to the rebuttal but the AC believes that most concerns are largely addressed.

---

### Decision · Program_Chairs · 2025-01-22

Accept (Poster)